# The signaling lipid sphingosine 1-phosphate regulates mechanical pain

Rose Z Hill[1], Benjamin U Hoffman[2,3], Takeshi Morita[1†], Stephanie M Campos[4‡], Ellen A Lumpkin[2,4], Rachel B Brem[5,6], Diana M Bautista[1,4,7]*

[1]Department of Molecular and Cell Biology, University of California, Berkeley, Berkeley, United States; [2]Department of Physiology and Cellular Biophysics, Columbia University College of Physicians and Surgeons, New York, United States; [3]Medical Scientist Training Program, Columbia University, New York, United States; [4]Neurobiology Course, Marine Biological Laboratory, Woods Hole, United States; [5]Department of Plant and Microbial Biology, University of California, Berkeley, Berkeley, United States; [6]Buck Institute for Research on Aging, Novato, United States; [7]Helen Wills Neuroscience Institute, University of California, Berkeley, Berkeley, United States

*For correspondence:
dbautista@berkeley.edu

Present address: †Rockefeller University, New York, United States; ‡Indiana University, Bloomington, United States

Competing interests: The authors declare that no competing interests exist.

**Abstract** Somatosensory neurons mediate responses to diverse mechanical stimuli, from innocuous touch to noxious pain. While recent studies have identified distinct populations of A mechanonociceptors (AMs) that are required for mechanical pain, the molecular underpinnings of mechanonociception remain unknown. Here, we show that the bioactive lipid sphingosine 1-phosphate (S1P) and S1P Receptor 3 (S1PR3) are critical regulators of acute mechanonociception. Genetic or pharmacological ablation of S1PR3, or blockade of S1P production, significantly impaired the behavioral response to noxious mechanical stimuli, with no effect on responses to innocuous touch or thermal stimuli. These effects are mediated by fast-conducting A mechanonociceptors, which displayed a significant decrease in mechanosensitivity in S1PR3 mutant mice. We show that S1PR3 signaling tunes mechanonociceptor excitability via modulation of KCNQ2/3 channels. Our findings define a new role for S1PR3 in regulating neuronal excitability and establish the importance of S1P/S1PR3 signaling in the setting of mechanical pain thresholds.
DOI: https://doi.org/10.7554/eLife.33285.001

## Introduction

Pain is a complex sensation. It serves to protect organisms from harmful stimuli, but can also become chronic and debilitating following tissue injury and disease. Distinct cells and molecules detect noxious thermal and mechanical stimuli. Thermal pain is detected by thermosensitive TRP channels in subsets of nociceptors (*Caterina et al., 2000*; *Vriens et al., 2011*), and gentle touch is detected by Piezo2 channels in low-threshold mechanoreceptors (LTMRs) (*Ranade et al., 2014*; *Woo et al., 2014*). Aδ high-threshold mechanoreceptors (HTMRs) have been shown to play a key role in responses to painful mechanical stimuli (*Arcourt et al., 2017*; *Ghitani et al., 2017*).

Recent studies have shown that there are at least two populations of HTMRs that mediate responses to noxious mechanical stimuli. The *Npy2r+* subpopulation of HTMRs mediates fast paw withdrawal responses to pinprick stimulation and terminates as free nerve endings in the epidermis (*Arcourt et al., 2017*). The *Calca+* subpopulation of circumferential-HTMRs responds to noxious force and hair pulling, and terminates as circumferential endings wrapped around guard hair follicles (*Ghitani et al., 2017*). Additionally, somatostatin-expressing interneurons of laminae I-III in the dorsal horn of the spinal cord receive input from nociceptors and are required for behavioral responses

to painful mechanical stimuli (*Duan et al., 2014*). Despite these advances in defining the cells and circuits of mechanical pain, little is known about the molecular signaling pathways in mechanonociceptors.

Here, we show that sphingosine 1-phosphate (S1P) is required for mechanical pain sensation. S1P is a bioactive lipid that signals via 5 G-protein coupled S1P Receptors (S1PRs 1–5). S1P signaling, mainly via S1PR1, plays a well-known role in immune cell migration and maturation (*Spiegel and Milstien, 2003*; *Matloubian et al., 2004*; *Schwab et al., 2005*). Additionally, recent studies have shown that S1PRs are expressed throughout the nervous system (*Janes et al., 2014*; *Mair et al., 2011*; *Camprubí-Robles et al., 2013*) and S1P signaling is associated with a variety of neuroinflammatory disorders, including multiple sclerosis (*Brinkmann et al., 2010*) and Alzheimer's disease (*Couttas et al., 2014*). S1P has been implicated in spontaneous pain (*Camprubí-Robles et al., 2013*) and thermal pain hypersensitivity (*Mair et al., 2011*; *Finley et al., 2013*; *Weth et al., 2015*), but due to conflicting accounts of S1P receptor expression in the CNS (*Janes et al., 2014*; *Weth-Malsch et al., 2016*) and PNS (*Mair et al., 2011*; *Camprubí-Robles et al., 2013*; *Usoskin et al., 2015*) as well as inconsistent reports on the effects of S1P on neuronal excitability (*Camprubí-Robles et al., 2013*; *Zhang et al., 2006*; *Li et al., 2015*) and pain behaviors (*Mair et al., 2011*; *Camprubí-Robles et al., 2013*; *Finley et al., 2013*; *Weth et al., 2015*), the role of S1P in somatosensation remains controversial.

We found that mice lacking the S1P receptor S1PR3 display striking and selective deficits in behavioral responses to noxious mechanical stimuli. Likewise, peripheral blockade of S1PR3 signaling or S1P production impairs mechanical sensitivity. We show that S1P constitutively enhances the excitability of A mechanonociceptors (AMs) via closure of KCNQ2/3 potassium channels to tune mechanical pain sensitivity. The effects of S1P are completely dependent on S1PR3. While previous studies have shown that elevated S1P triggers acute pain and injury-evoked thermal sensitization (*Mair et al., 2011*; *Camprubí-Robles et al., 2013*), we now demonstrate that baseline levels of S1P are necessary and sufficient for setting normal mechanical pain thresholds. By contrast, elevated S1P selectively triggers thermal sensitization via activation of TRPV1$^+$ heat nociceptors, with no effect on mechanical hypersensitivity. Our findings uncover an essential role for constitutive S1P signaling in mechanical pain.

## Results

To identify candidate genes underlying mechanosensation, we previously performed transcriptome analysis of the sensory ganglia innervating the ultra-sensitive tactile organ (the star) of the star-nosed mole (*Gerhold et al., 2013*). Immunostaining revealed the tactile organ is preferentially innervated by myelinated Aδ fibers (*Gerhold et al., 2013*), which are primarily mechanosensitive. While our original analysis focused on ion channels enriched in the neurons of the star organ, our dataset also revealed enrichment of several components of the S1P pathway, including *S1pr3*. Likewise, single-cell RNA seq of mouse dorsal root ganglion (DRG) neurons revealed *S1pr3* expression in a subset of myelinated mechanoreceptors (*Usoskin et al., 2015*) in addition to a subpopulation of peptidergic C nociceptors.

S1P promotes excitability in small-diameter, capsaicin-sensitive nociceptors (*Mair et al., 2011*; *Camprubí-Robles et al., 2013*; *Zhang et al., 2006*; *Li et al., 2015*). In addition, S1PR3 has been shown to mediate spontaneous pain triggered by elevated S1P and thermal sensitization following sterile tissue injury (*Camprubí-Robles et al., 2013*). However, no studies have examined the role of S1PR3 in mechanosensation or in regulating somatosensory behaviors under normal conditions. Given the enrichment of *S1pr3* in mechanosensory neurons of the star-nosed mole and mouse, we hypothesized that S1P signaling via S1PR3 may also play a role in mechanosensation. Thus, we set out to define the role of S1P signaling and S1PR3 in somatosensory mechanoreceptors.

### S1PR3 mediates acute mechanical pain

We first examined a variety of somatosensory behaviors in mice lacking S1PR3 (*Kono et al., 2004*) (*S1pr3$^{tm1Rlp/Mmnc}$*; referred to herein as S1PR3 KO). We initially investigated baseline responses to mechanical stimuli. S1PR3 KO mice displayed a dramatic loss of mechanical sensitivity (*Figure 1A*; see *Figure 1—source data 1*), as von Frey paw withdrawal thresholds were significantly elevated in S1PR3 KO mice relative to WT and S1PR3 HET littermates (mean thresholds: 1.737 g vs. 0.736 and

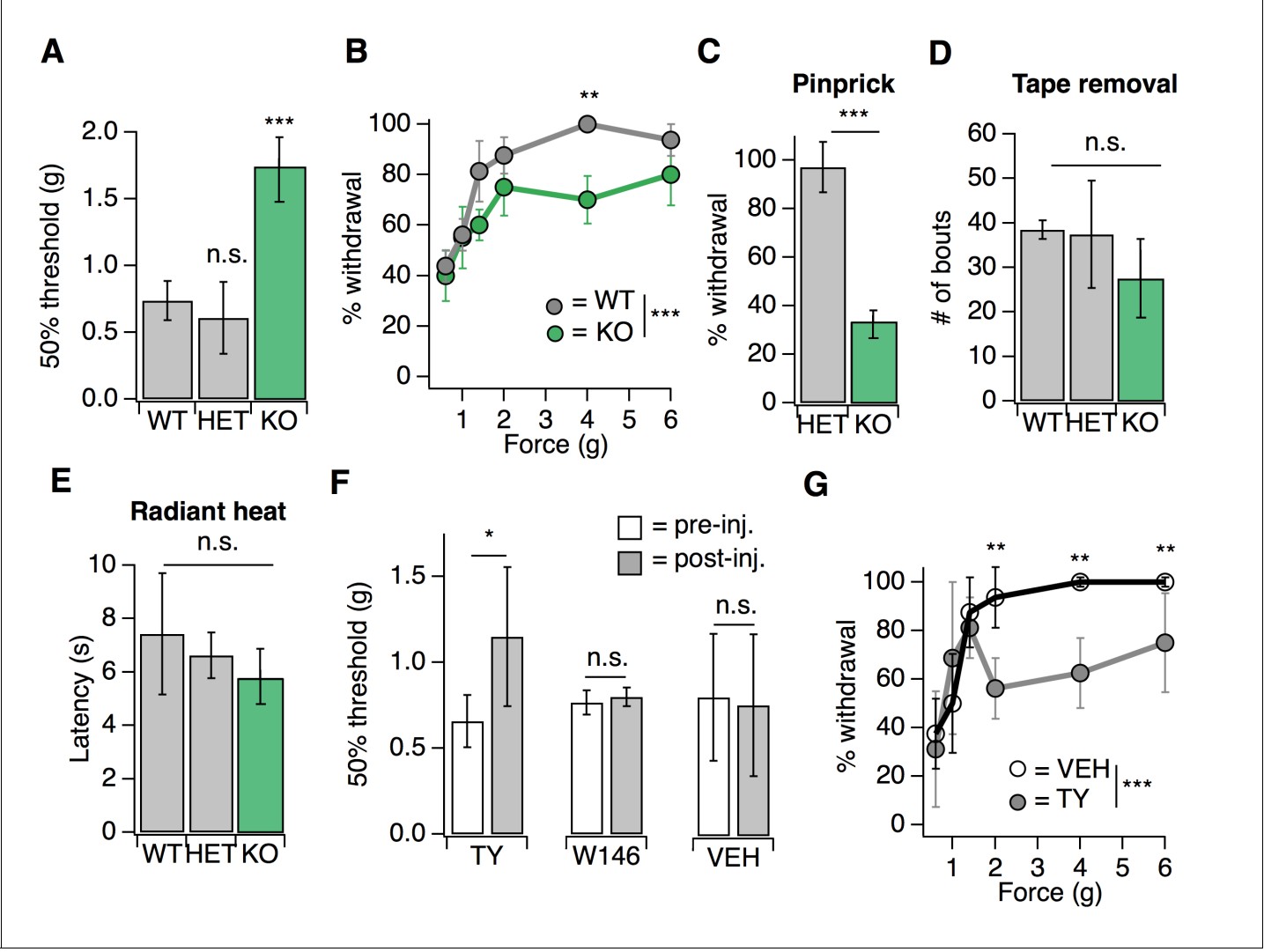

**Figure 1.** S1PR3 mediates acute mechanical pain. (A) von Frey 50% withdrawal threshold measurements for $S1pr3^{+/+}$ (WT, N = 8), $S1pr3^{+/-}$ (HET, N = 7) and $S1pr3^{-/-}$ (KO, N = 12) mice. p<0.0001 (one-way ANOVA). Tukey-Kramer post hoc comparisons for KO and HET to WT indicated on graph. (B) von Frey force-response graph for WT (N = 8) versus KO (N = 12) animals; $p_{genotype}$ <0.0001 (two-way ANOVA). Tukey HSD comparisons between genotypes are indicated for given forces. (C) % withdrawal to pinprick stimulation of hindpaw for HET versus KO animals; p<0.0001 (unpaired t-test; N = 5–7 mice per group). (D) Number of attempted removal bouts in tape assay for WT (N = 2), HET (N = 2), and KO (N = 5) mice; p=0.172 (one-way ANOVA). (E) Baseline radiant heat measurements for WT (N = 8), HET (N = 3), and KO (N = 5) mice; p=0.444 (one-way ANOVA). (F) von Frey 50% withdrawal threshold measurements for mice pre- and post-injection of 500 µM TY 52156 (N = 10), 10 µM W146 (N = 6), or 1% DMSO-PBS vehicle (N = 17); p=0.016, 0.650 (two-tailed paired t-test comparing vehicle- vs. drug-injected paw). (G) von Frey force-response graph for mice injected with either 1% DMSO-PBS (N = 4) or 500 µM TY 52156 (N = 4); $p_{treatment}$ <0.0001 (two-way ANOVA). Tukey HSD comparisons were made between treatment groups and significant differences at a given force are indicated on graph. Error bars represent mean ± SD.
DOI: https://doi.org/10.7554/eLife.33285.002
The following source data and figure supplement are available for figure 1:

**Source data 1.** S1PR3 mediates acute mechanical pain.
DOI: https://doi.org/10.7554/eLife.33285.004
**Figure supplement 1.** Loss of S1PR3 selectively impairs mechanonociception.
DOI: https://doi.org/10.7554/eLife.33285.003

0.610 g, respectively). Moreover, S1PR3 KO mice demonstrated decreased responses to a range of noxious tactile stimuli (2–6 g; *Figure 1B*) and to noxious pinprick stimulation (*Figure 1C*), but normal responsiveness to innocuous tactile stimuli (0.6–1.4 g; *Figure 1B*). S1PR3 KO mice exhibited normal tape removal attempts (*Ranade et al., 2014*) (*Figure 1D*), righting reflexes (*Figure 1—figure*

supplement 1A), radiant heat withdrawal latencies (Figure 1E), and itch-evoked scratching (Figure 1—figure supplement 1B). These results demonstrate a selective role for S1PR3 in acute mechanical pain.

As a complement to our analysis of somatosensation in S1PR3 KO animals, we employed a pharmacological approach, using the S1PR3-selective antagonist TY 52156 (TY) (Nussbaum et al., 2015). Similar to the phenotype of knockout animals, intradermal injection of 500 µM TY into the mouse hindpaw (the site of testing) triggered a rapid and significant elevation in von Frey paw withdrawal thresholds (Figure 1F) and decreased responsiveness to noxious (2–6 g), but not innocuous (0.6–1.4 g), tactile stimuli (Figure 1G), without affecting noxious heat sensitivity (Figure 1—figure supplement 1C). By contrast, blockade of S1PR1 with the selective antagonist W146 (Finley et al., 2013) had no effect on baseline mechanical or thermal thresholds (Figure 1F; Figure 1—figure supplement 1C). Overall, these data show that S1PR3 signaling sets mechanical pain sensitivity.

## Endogenous S1P mediates acute mechanical pain

We next asked whether peripheral S1P was required for the S1PR3-dependent effects on mechanosensation. We decreased S1P levels via injection of the sphingosine kinase inhibitor SKI II to block local production of S1P (Chiba et al., 2010) or elevated S1P levels via intradermal injection of S1P and measured behaviors 30 min after injection. Decreasing local S1P levels with SKI II significantly reduced mechanical sensitivity (Figure 2A; see Figure 2—source data 1), comparable to the hyposensitivity phenotype observed in S1PR3 KO mice (Figure 1A). Again, similar to what was observed in S1PR3 KO animals (Figure 1E), peripheral blockade of S1P production had no effect on baseline thermal sensitivity (Figure 1—figure supplement 1C). Surprisingly, injecting exogenous S1P (10 µM; maximum solubility in saline vehicle) had no effect on mechanical sensitivity (Figure 2A–B). However, as previously reported (Mair et al., 2011; Camprubí-Robles et al., 2013), S1P injection triggered S1PR3-dependent thermal hypersensitivity and spontaneous pain (Figure 2C–D), demonstrating that the lack of effect on mechanical hypersensitivity is not due to problems with S1P delivery or degradation.

These data support a model whereby S1P constitutively activates S1PR3 to set normal mechanical pain thresholds. To further test this model, we asked if the mechanical hyposensitivity elicited after endogenous S1P depletion (via SKI II) could be rescued by local injection of exogenous S1P. Indeed, we found that injection of exogenous S1P reversed SKI II-induced mechanical hyposensitivity in a dose-dependent manner, and observed a maximal effect with 200 nM S1P (Figure 2E). Although quantification of native S1P levels in skin is inaccurate owing to avid lyase activity (Shaner et al., 2009), our data establish that baseline S1P levels are sufficient to maximally exert their effect on S1PR3-dependent mechanical pain, such that increased S1P does not evoke mechanical hypersensitivity, but diminished S1P leads to mechanical hyposensitivity. These data show that constitutive activation of S1PR3 by S1P is required for normal mechanosensitivity.

## S1PR3 is expressed in A mechanonociceptors and thermal nociceptors

Our behavioral data showing distinct roles for S1PR3 in mechanonociception and thermal hypersensitivity suggest that S1PR3 is expressed in distinct subsets of somatosensory neurons. While a previous study suggested that all somatosensory neurons express S1PR3 (Camprubí-Robles et al., 2013), single cell RNA seq data suggests S1pr3 is not expressed by all DRG neurons (Usoskin et al., 2015), and no studies have performed quantitative analysis of S1PR3 staining or co-staining to define subpopulations of S1PR3[+] neurons. We thus set out to characterize the somatosensory neuron subtypes expressing S1pr3 using in situ hybridization (ISH) of wild-type somatosensory ganglia and immunohistochemistry (IHC) in an S1pr3[mCherry/+] reporter mouse (Sanna et al., 2016).

We first used in situ hybridization (ISH) with a specific S1pr3 probe to examine expression patterns of S1pr3 (Figure 3A–B; see Supplementary file 1). In our experiments, 43% of cells from wild-type DRG expressed S1pr3. Co-ISH revealed that one population of S1pr3[+] neurons represents Aδ mechanonociceptors (AMs). These cells expressed Scn1a (39.9% of all S1pr3[+]), a gene that encodes the Nav1.1 sodium channel, which mediates mechanical pain in Aδ fibers (Osteen et al., 2016). S1pr3[+] cells also co-expressed Npy2r (20.4% of all S1pr3[+]), a marker of a subset of mechanonociceptive A fibers (Arcourt et al., 2017). S1pr3 was expressed in 70.6% of Scn1a[+] cells and 72% of Npy2r[+] cells, comprising a majority of both of these populations. Interestingly, a subset of cells co-

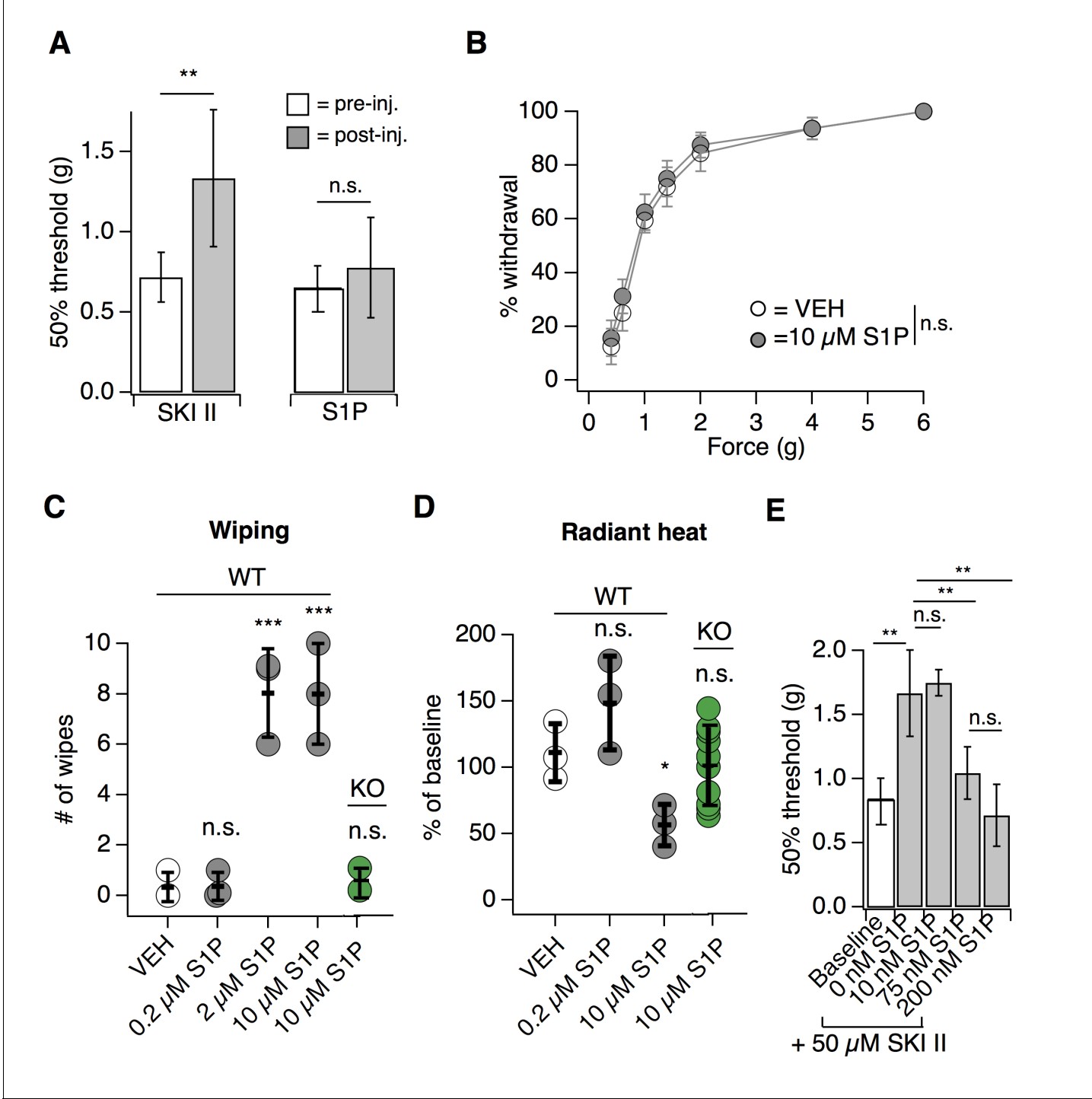

**Figure 2.** Endogenous S1P mediates acute mechanical pain. (**A**) von Frey 50% withdrawal measurements for mice pre- and post-injection of 50 μM SKI II (N = 8) or 10 μM S1P (N = 7); p=0.003, 0.604 (two-tailed paired t-tests). (**B**) von Frey force-response graph for animals injected with 10 μM S1P or 0.1% MeOH-PBS; $p_{genotype}$ >0.05 (two-way ANOVA; N = 8 mice per group). No Tukey HSD comparisons at any force between genotypes were significant. (**C**) Intradermal cheek injection of 10 μM S1P, 2 μM, 0.2 μM, and 20 μL 0.3% methanol PBS (vehicle), with quantification of number of forepaw wipes over the 5 min post-injection interval; p<0.0001 (one-way ANOVA; N = 3 mice per condition). Dunnett's multiple comparisons p-values are represented on graph for comparisons made between treated and vehicle groups. (**D**) Radiant heat normalized paw withdrawal latencies 20–30 min post injection of 15 μL 10 μM S1P, 0.2 μM S1P. or 0.3% methanol-PBS vehicle (i.d.) into the hind paw of S1PR3 WT or KO mice; p=0.0129 (one-way ANOVA; N = 3–10 mice per condition). Dunnett's multiple comparisons p-values are represented on graph for comparisons made between treated and vehicle groups. (**E**) von

*Figure 2 continued on next page*

*Figure 2 continued*

Frey 50% withdrawal measurements for mice pre- (baseline) and post-injection of 50 μM SKI II (N = 14) and 0 (N = 4), 10 (N = 3), 75 (N = 4), or 200 nM S1P (N = 3; one-way ANOVA; *p*=0.0001). Tukey Kramer comparisons are indicated on graph. Error bars represent mean ± SD.

DOI: https://doi.org/10.7554/eLife.33285.005

The following source data is available for figure 2:

**Source data 1.** Endogenous S1P mediates acute mechanical pain.

DOI: https://doi.org/10.7554/eLife.33285.006

expressed *S1pr3* and the mechanically sensitive channel *Piezo2*, which is expressed by Aβ, Aδ, and C fibers (*Ranade et al., 2014*). The remaining *S1pr3*[+] cells were *Trpv1*[+] and/or *Trpa1*[+] C nociceptors (67.1% of all *S1pr3*[+]), which are reported to overlap minimally with the *Scn1a*[+] and *Npy2r*[+] populations (*Arcourt et al., 2017*; *Osteen et al., 2016*).

We next used an *S1pr3*[mCherry/+] reporter mouse, which produces a functional S1PR3-mCherry fusion protein (*Sanna et al., 2016*), as an independent strategy to explore S1PR3 expression and localization. This strategy was used because we found that anti-S1PR3 antibodies showed broad immunoreactivity in DRG from mice lacking S1PR3, and so we instead used anti-DsRed antibodies to probe expression of the S1PR3 fusion protein (*Figure 3—figure supplement 1E*). We found that 42.4% of S1PR3[+] cells co-stained with anti-Peripherin, demonstrating that S1PR3 is expressed in a subset of small-diameter neurons. We also observed that 69.5% of S1PR3[+] cells co-stained with anti-NF200, which marks medium and large-diameter myelinated neurons. Furthermore, we observed that S1PR3[+] cells were primarily of small to medium diameter (11.3–35.1 μm), whereas all cells in the DRG ranged from 11.3 to 53.9 μm. Overall, these data support the expression of S1PR3 in subsets of small-diameter thermal nociceptors and medium-diameter mechanonociceptors (*Figure 3F*). Additionally, no significant differences were observed between WT and S1PR3 KO DRG in number of *Trpa1*[+], *Trpv1*[+], Peripherin[+], NF200[+], or IB4[+] cells (*Figure 3—figure supplement 1B–C,F,G*). The mean diameters of *Trpv1*[+] neurons (*Figure 3—figure supplement 1D*, left), NF200[+] neurons (*Figure 3—figure supplement 1G*), or all neurons (*Figure 3—figure supplement 1D*, right) in WT versus KO DRG were not significantly different, suggesting no loss of major sensory neuronal subtypes in the S1PR3 KO.

We then visualized S1PR3 expression in nerve fibers that innervate the skin using anti-DsRed antibodies in whole-mount immunohistochemistry (IHC; *Figure 3D*). The reporter animals showed no specific antibody staining in epidermal or dermal cells (*Figure 3—figure supplement 1I*), and single-cell RNA seq of a diverse array of mouse epidermal and dermal cells corroborates this lack of expression (*Joost et al., 2016*). We observed overlap of S1PR3-expressing free nerve endings with NF200[+] myelinated free nerves and NF200- putative C-fiber endings (*Figure 3F*), but did not observe expression of S1PR3 in NF200[+] circumferential or lanceolate hair follicle receptors, or in putative Merkel afferents (*Figure 3D–E*). β-tubulin III, PGP9.5 (pan-neuronal markers), and NF200 staining in S1PR3 KO skin displayed patterns of epidermal and dermal innervation similar to WT skin, suggesting the phenotypes observed in the S1PR3 KO mice are not due to developmental loss of sensory neuronal innervation ($p_{PGP9.5}$= 0.443 (n = 93, 38 fibers); $p_{NefH}$ = 0.405 (n = 61, 28 fibers); $p_{BTIII}$ = 0.353 (n = 104, 89 fibers); two-tailed t-tests based on average number of fibers per field of view). These results support expression of S1PR3 in subsets of myelinated A mechanonociceptors and unmyelinated C nociceptors that terminate as free nerve endings.

## S1P activates thermal nociceptors but not putative AMs

Live imaging of cultured DRG neurons from adult reporter animals showed expression of S1PR3-mCherry fusion protein in 48.3% of neurons, mirroring our ISH and IHC results (*Figure 4A*). To examine the effects of S1P on A mechanonociceptors and C nociceptors, we performed ratiometric calcium imaging and electrophysiology on DRG cultures from reporter mice. Interestingly, only 56.1 ± 22.4% of mCherry-expressing neurons were activated by 1 μM S1P (Representative trace in *Figure 4B*; representative images in *Figure 4C*), which our dose-response showed to be the saturating concentration for calcium influx (*Figure 4D*; $EC_{50}$ = 155 nM). All S1P-responsive neurons were also capsaicin-sensitive (*n* > 2000 neurons). And while sensory neurons from S1PR3 KO animals did not respond to S1P, as expected (*Camprubí-Robles et al., 2013*), they exhibited capsaicin responses

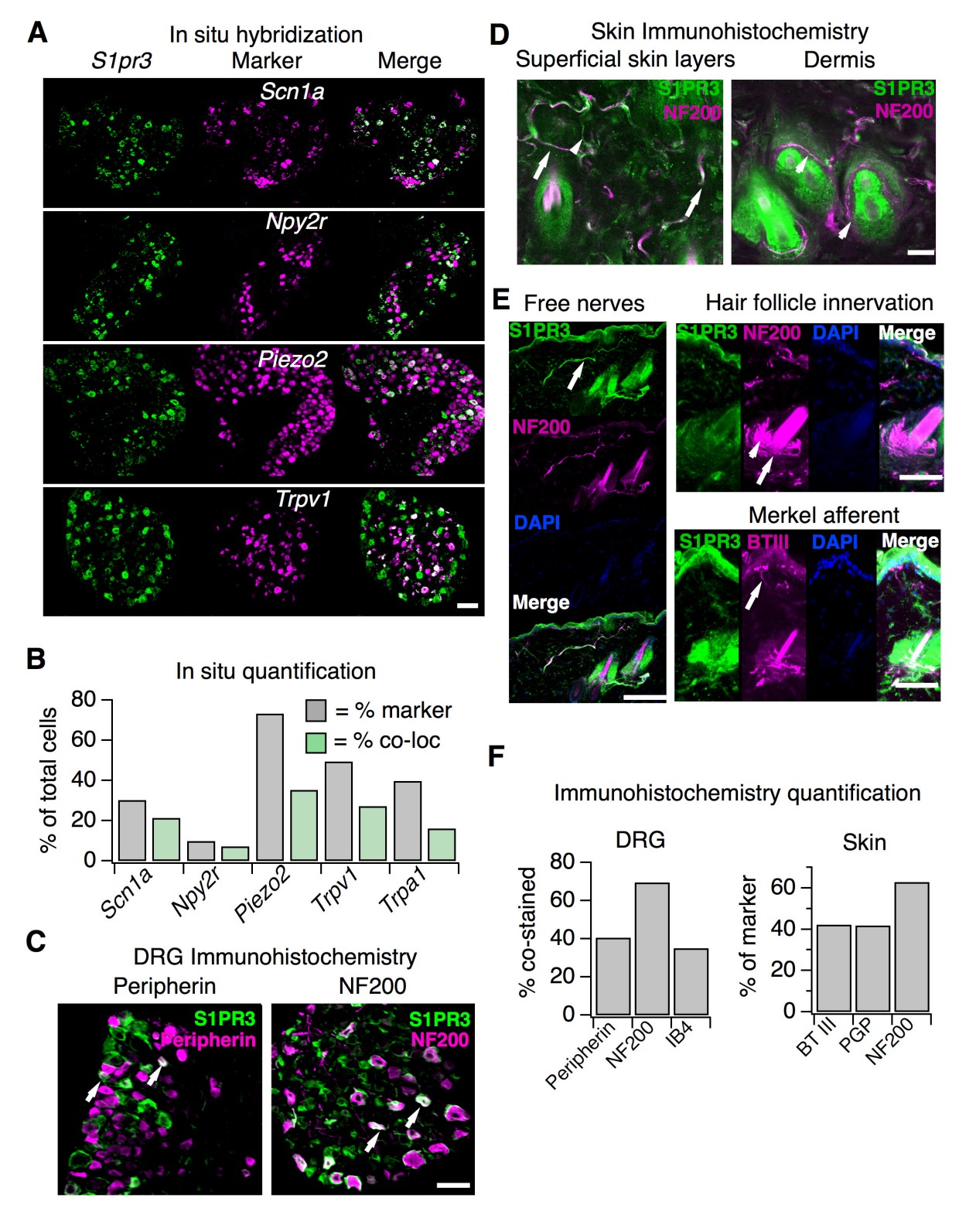

**Figure 3.** *S1pr3* is expressed in A mechanonociceptors and C thermal nociceptors. (**A**) (Top) Representative co-ISH of *S1pr3* (green; left) with *Scn1a*, *Npy2r*, *Piezo2*, and *Trpv1* (magenta; center) in sectioned DRG. Right column: overlay with co-localized regions colored white (10x air objective; scale = 100 μm). (**B**) Bar chart showing the % of total cells expressing the indicated marker (grey) and the % of total cells co-expressing both marker and *S1pr3* (green). See Table S1 for quantification. (**C**) Representative IHC images of sectioned DRG from *S1pr3^{mCherry/+}* animals stained with anti-DsRed

*Figure 3 continued on next page*

*Figure 3 continued*
(green, S1PR3) and anti-Peripherin (left, magenta) or anti-NF200 (right, magenta). Arrows indicate co-stained cells. Images were acquired using a 10x air objective (scale = 100 µm). (D) Whole-mount skin IHC confocal images with anti-DsRed antibody (S1PR3, green) and anti-NefH antibody (NF200, magenta) in an *S1pr3*^mCherry/+^ animal (20x water objective; scale = 50 µm). Arrows indicate co-positive free nerves (left image). Arrowheads indicate NF200- free nerves (left) or S1PR3- circumferential fibers (right image). (E) Sectioned skin IHC with anti-DsRed (S1PR3) and anti-NefH (NF200, left, top right) or anti-DsRed (S1PR3) and anti-beta-tubulin III (BTIII, bottom right) antibody (magenta) in *S1pr3*^mCherry/+^ skin (20x air objective; scale = 50 µm). Arrows indicate co-positive free nerve endings (left), S1PR3-negative lanceolate/circumferential hair follicle endings (top right, arrow = circumferential, arrowhead = lanceolate), or S1PR3-negative putative Merkel afferent (bottom right). (F) (Left) Quantification of sectioned DRG IHC experiments showing % of S1PR3+ cells that co-stained with indicated markers (n > 250 cells per marker). (Right) Quantification of sectioned skin IHC experiments showing % of fibers positive for indicated marker that co-stained with S1PR3 (anti-DsRed; n = 10 images per marker from two animals).
DOI: https://doi.org/10.7554/eLife.33285.007
The following figure supplement is available for figure 3:

**Figure supplement 1.** S1PR3 KO animals display normal representation of somatosensory neuronal subtypes.
DOI: https://doi.org/10.7554/eLife.33285.008

that were not significantly different from WT neurons (*Figure 3—figure supplement 1H*). The mean diameter of S1P-responsive mCherry$^+$ neurons was 22.4 ± 1.0 µm, whereas the mean diameter of non-responsive mCherry$^+$ neurons was 28.7 ± 3.2 µm (p=0.0002, two-tailed t-test). We also performed whole cell current clamp experiments and, consistent with other studies (*Mair et al., 2011*; *Zhang et al., 2006*; *Li et al., 2015*), found that S1P evoked action potential firing in capsaicin-sensitive small diameter cells (*Figure 4E*). This shows that only the small-diameter, S1PR3$^+$ putative nociceptors are excited by S1P. We next asked whether the S1PR3$^+$ medium-large diameter neurons represent the mechanonociceptors observed by ISH (*Figure 3A*). To this end, we asked whether the spider toxin Hm1a, a selective activator of AM nociceptors (*Osteen et al., 2016*), triggers calcium influx in S1PR3-expressing trigeminal neurons. Indeed, we found that 44.2 ± 15.1% of Hm1a-responsive neurons expressed mCherry (*Figure 4F*), consistent with our staining showing expression of *S1pr3* in AM nociceptors and the role of Hm1a-responsive neurons in mediating mechanical pain in vivo (*Osteen et al., 2016*).

## S1PR3 modulates KCNQ2/3 channels to regulate AM excitability

We next interrogated the molecular mechanism by which S1P signaling in AM nociceptors may regulate mechanical pain. We performed whole-cell current clamp on the medium-diameter *S1pr3*^mCherry/+^ dissociated DRG neurons (membrane capacitance = 61.05 ± 1.92 pF), which did not display S1P-evoked calcium influx (*Figure 4B–C*). In these cells, 1 µM S1P application did not change membrane potential (*Figure 5—figure supplement 1A*; see *Figure 5—source data 1*) or elicit firing in the absence of current injection (*Figure 5—figure supplement 1A*; *Figure 5A*). However, S1P dramatically lowered the threshold to fire action potentials (rheobase) in an S1PR3-dependent manner (*Figure 5A*, *Figure 5—figure supplement 1B*).

We then set out to determine the mechanism by which S1PR3 activity increases neuronal excitability using whole-cell voltage clamp recording. Previous studies showed that S1P excites capsaicin-sensitive nociceptors by increasing voltage-gated sodium currents and reducing steady-state potassium currents (*Zhang et al., 2006*; *Li et al., 2015*). We found that S1P had no such effects on S1PR3$^+$ medium-diameter cells (*Figure 5—figure supplement 1C–E*). By contrast, S1P triggered a robust increase in input resistance (*Figure 5B*), consistent with the closure of potassium channels. I-V analysis revealed that the current inhibited by S1P application was carried by potassium (*Figure 5C*). Additionally, S1P significantly reduced slow, voltage-dependent tail current amplitudes (*Figure 5—figure supplement 1F*; *Figure 5D* (top)) in an S1PR3-dependent manner (*Figure 5—figure supplement 1F*, center).

As tail currents in Aδ neurons are primarily mediated by KCNQ2/3 potassium channels (*Schütze et al., 2016*; *Passmore et al., 2012*), we postulated that S1P may alter tail currents through modulation of these channels. Furthermore, the above properties of the S1P-sensitive current were consistent with the reported electrophysiological properties of KCNQ2/3 channels in DRG neurons (*Schütze et al., 2016*; *Crozier et al., 2007*; *Xu et al., 2010*). To address whether KCNQ2/3 channels mediated S1P-dependent neuronal excitability, we applied the KCNQ2/3-selective inhibitor XE 991 and found that it completely occluded the effects of S1P on tail current (*Figure 5D*). Similar

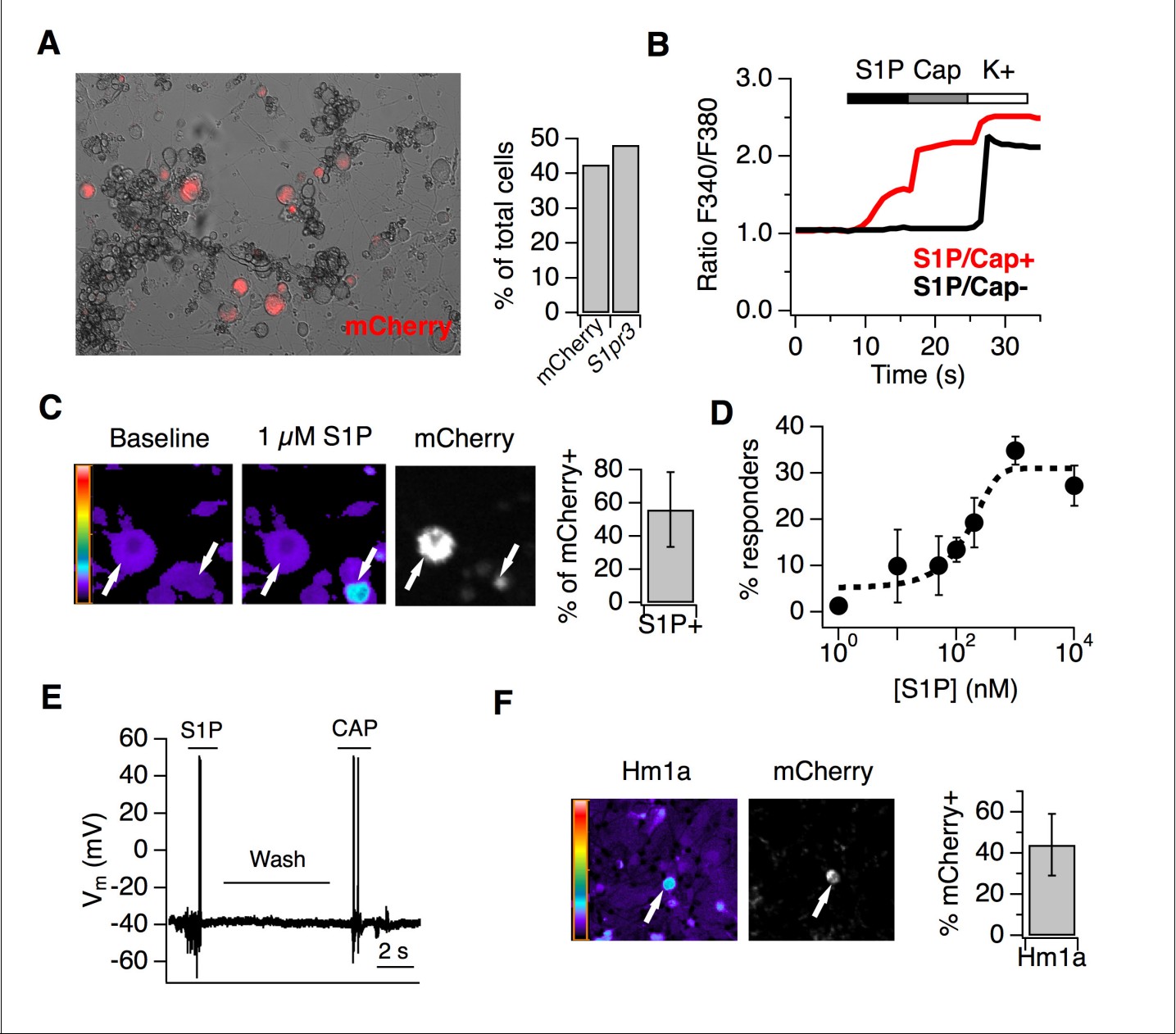

**Figure 4.** S1P activates thermal nociceptors but not mechanonociceptors. (A) (Left) Representative image of mCherry signal in live, cultured adult DRG neurons from one *S1pr3*$^{mCherry/+}$ animal. (Right) Quantification of % of total cells expressing *S1pr3* from DRG ISH and mCherry from dissociated DRG cultures (N = 2 animals each experiment). (B) Representative traces depicting F340/F380 signal from Fura2-AM calcium imaging showing two neurons, one which responded to 1 μM S1P, 1 μM Capsaicin, and high K + Ringer's (red) and one which only responded to high K+ (black). (C) (Left) Fura-2 AM calcium imaging before (left) and after (center) addition of 1 μM S1P in *S1pr3*$^{mCherry/+}$ cultured mouse DRG neurons. Bar indicates fluorescence ratio. Right-hand image indicates mCherry fluorescence. (Right) % of mCherry neurons that are responsive to 1 μM S1P in ratiometric calcium imaging (n > 1000 cells from 16 imaging wells from three animals). (D) Dose-response curve of mean neuronal calcium responders to varying concentrations of S1P. Concentrations used: 1, 10, 50, 100, 200, 1000, and 10,000 nanomolar (N = 2 animals). Error bars represent mean ± SD. Black dotted line indicates sigmoidal fit for all S1P responders from which EC$_{50}$ was derived. All S1P responders were also capsaicin-responsive. (E) Current-clamp trace of a single wild-type neuron firing action potentials in response to bath addition of 1 μM S1P and 1 μM capsaicin, with Ringer's wash in-between. Four of tenneurons responded to S1P and one of one S1P-responsive also responded to capsaicin. Bar = 2 s. (F) (Left) Fura-2 AM calcium imaging after addition of 500 nM Hm1a in *S1pr3*$^{mCherry/+}$ P0 TG neurons, which were used instead of adult DRG neurons because they respond to Hm1a without prior PGE$_2$ sensitization. Right-hand image indicates mCherry fluorescence. (Right) % of Hm1a-responsive P0 TG neurons that are mCherry+ (N = 1 animal, 1230 total neurons).

DOI: https://doi.org/10.7554/eLife.33285.009

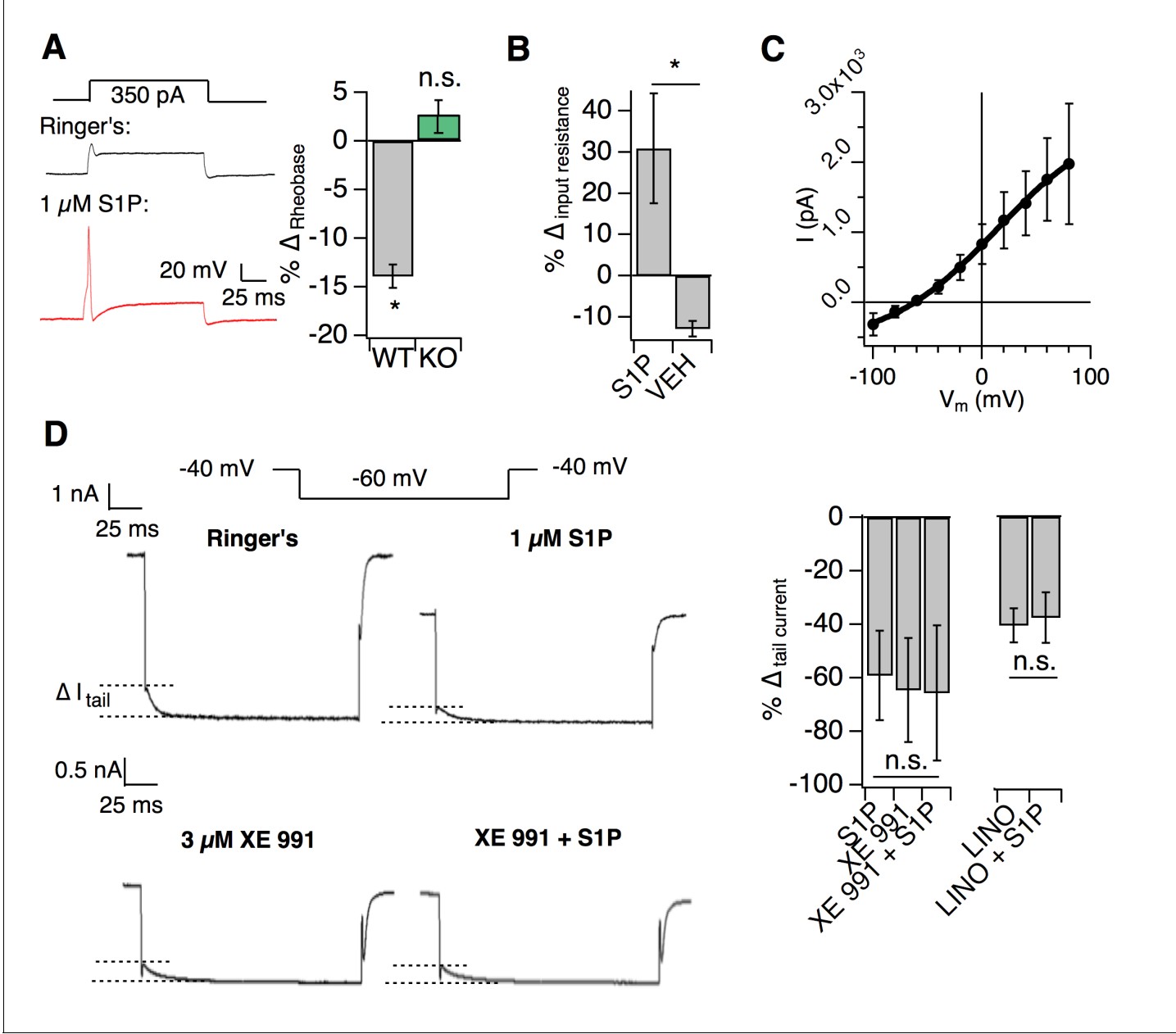

**Figure 5.** S1PR3 modulates KCNQ2/3 channels to regulate AM excitability. All experiments were performed in *S1pr3*^mCherry/+ or ^-/- DRG neurons. (**A**) (Left) Example traces of a single mCherry +neuron in whole cell current clamp before and after S1P application. (Right) % change in rheobase after S1P application for *S1pr3*^mCherry/+ (left, n = 7) and KO (right, n = 12) neurons ($p_{WT,KO}$ = 0.012, 0.287; two-tailed paired t-tests). (**B**) % Δ in input resistance after S1P or vehicle application (p=0.017; two-tailed paired t-test; n = 4 cells per group). (**C**) The S1P-sensitive current is carried by potassium. The current-voltage relationship was determined by subtraction of the post-S1P current from the pre-S1P current and reverses at −60.125 mV; n = 6 cells. Data were fitted with a Boltzmann equation. Pre- and post-S1P currents were measured at the indicated voltage (−100 mV to +80 mV, 20 mV increments) following a +100 mV step (100 ms). Current was quantified using the peak absolute value of the slowly-deactivating current 0–10 ms after stepping to indicated voltage. Unless indicated otherwise, all error bars represent mean ± SEM. (**D**) (Graphic, top) Averaged current traces of a single mCherry+ neuron in whole cell voltage clamp recording comparing tail currents ($\Delta_{I\ tail}$) pre- and post-S1P using indicated voltage step protocol. (graphic, bottom) Averaged current traces of a single mCherry+ neuron in whole cell voltage clamp recording with XE991 treatment. Holding phase (−40 mV, 150 ms) was truncated in traces. (Left graph) % Δ in outward tail current (average ±SD after indicated treatments (1 μM S1P, 3 μM XE 991, or both) for *S1pr3*^mCherry/+ medium-diameter neurons; (p=0.58; one-way ANOVA; n = 6, 8, 14 cells) using protocol depicted at right. (Right graph) % Δ in inward tail current after indicated treatments (LINO = 100 μM linopirdine) for *S1pr3*^mCherry/+ medium-diameter neurons; (p=0.47; two-tailed paired t-test; n = 12 cells).

DOI: https://doi.org/10.7554/eLife.33285.010

*Figure 5 continued on next page*

*Figure 5 continued*

The following source data and figure supplement are available for figure 5:

**Source data 1.** S1PR3 modulates KCNQ2/3 channels to regulate AM excitability.

DOI: https://doi.org/10.7554/eLife.33285.012

**Figure supplement 1.** S1P selectively modulates potassium tail currents to increase DRG neuron excitability.

DOI: https://doi.org/10.7554/eLife.33285.011

results were observed with the related antagonist, linopirdine (*Figure 5D*). These findings are consistent with S1P/S1PR3-dependent inhibition of KCNQ2/3 in somatosensory neurons.

We also found that the effect of S1P on KCNQ2/3 currents was mediated by low levels of S1P, exhibiting an $IC_{50}$ of 48 nM with saturation at 100 nM (*Figure 5—figure supplement 1G*). While S1P cannot be accurately measured in non-plasma tissues, this is similar to estimated levels of S1P in peripheral tissues (*Schwab et al., 2005*; *Ramos-Perez et al., 2015*), and to levels which rescued mechanosensitivity after local S1P depletion (*Figure 2E*). Thus, our in vitro $IC_{50}$ supports our finding that baseline S1P levels are sufficient to maximally exert their effect on mechanical pain. In summary, our electrophysiological and behavioral observations support a model in which baseline S1P/S1PR3 signaling governs mechanical pain thresholds through modulation of KCNQ2/3 channel activity in AM neurons.

## S1PR3 is required for nociceptive responses of high-threshold AM nociceptors

Given the effects of S1P on putative AM neurons in vitro and the selective attenuation of baseline mechanical pain in S1PR3 KO animals, we hypothesized that S1PR3 would play a role in AM afferent function. To test this, we utilized ex vivo skin-nerve recordings to analyze the effects of genetic ablation of S1PR3 on AM afferents, which mediate fast mechanical pain sensation. S1PR3 HET animals were used as littermate controls because no significant differences were observed between S1PR3 WT and S1PR3 HET mice in any behavioral assay (*Figure 1*), and because force-response relationships are comparable between S1PR3 HET AM fibers and wild type AM recordings (*Osteen et al., 2016*; *McIlwrath et al., 2007*; *Kwan et al., 2009*; *Smith et al., 2013*; *Garrison et al., 2012*) (*Figure 6—figure supplement 1A*; see *Figure 6—source data 1*). Compared to S1PR3 HET, S1PR3 KO AM nociceptors displayed reduced sensitivity in their force-response relation (slope for HET versus KO: 50 Hz/N versus 35 Hz/N), as well as attenuated firing over the noxious, but not innocuous, range of mechanical stimulation (*Figure 6A*). Strikingly, the median von Frey threshold to elicit firing in AM nociceptors was significantly higher in S1PR3 KO animals (3.92 mN) compared to littermate controls (1.56 mN; *Figure 6B*). Furthermore, S1PR3 KO AM nociceptors displayed a right-shifted cumulative response curve to force-controlled stimuli (50% effective force for HET versus KO: 33.7 versus 60.0 mN; *Figure 6C*), consistent with the mechanonociceptive hyposensitivity observed in vivo. By contrast, neither AM conduction velocities nor the conduction velocity distributions of Aβ, Aδ, and C fibers differed between genotypes (*Figure 6D* and *Figure 6—figure supplement 1B*).

A recent study reported that A-nociceptors are composed of two genetically distinct neuronal populations that differ in conduction velocity and in adaptation properties (*Arcourt et al., 2017*) ('Adapting AM' versus 'Non-adapting AM'). We next asked whether loss of S1PR3 signaling altered these AM subtypes. Adapting AM fibers responded more vigorously to dynamic (ramp) stimuli than to static (hold) stimuli, and displayed a mean dynamic firing frequency at least twofold greater than their static firing frequency (*Arcourt et al., 2017*) (*Figure 6E*, upper traces). By contrast, Non-adapting AM fibers often showed bursting during static stimulation, which resulted in similar firing rates during dynamic and static stimulation (*Figure 6E*, lower traces). S1PR3 KO animals displayed a significantly lower proportion of Adapting AM nociceptors compared with littermate controls (*Figure 6F*). Additionally, we observed an increase in S1PR3 KO AM fibers that were unresponsive to controlled force stimulation (*Figure 6F*). These 'non-responders' only fired action potentials to high-pressure stimuli with a blunt glass probe or to suprathreshold stimulation with von Frey filaments (see Methods). The Non-adapting AMs, and the few remaining mechanosensitive Adapting AMs in the S1PR3 KO displayed similar firing frequencies over both the dynamic and static phases of force application to control fibers (*Figure 6—figure supplement 1C*). This suggests that decreased

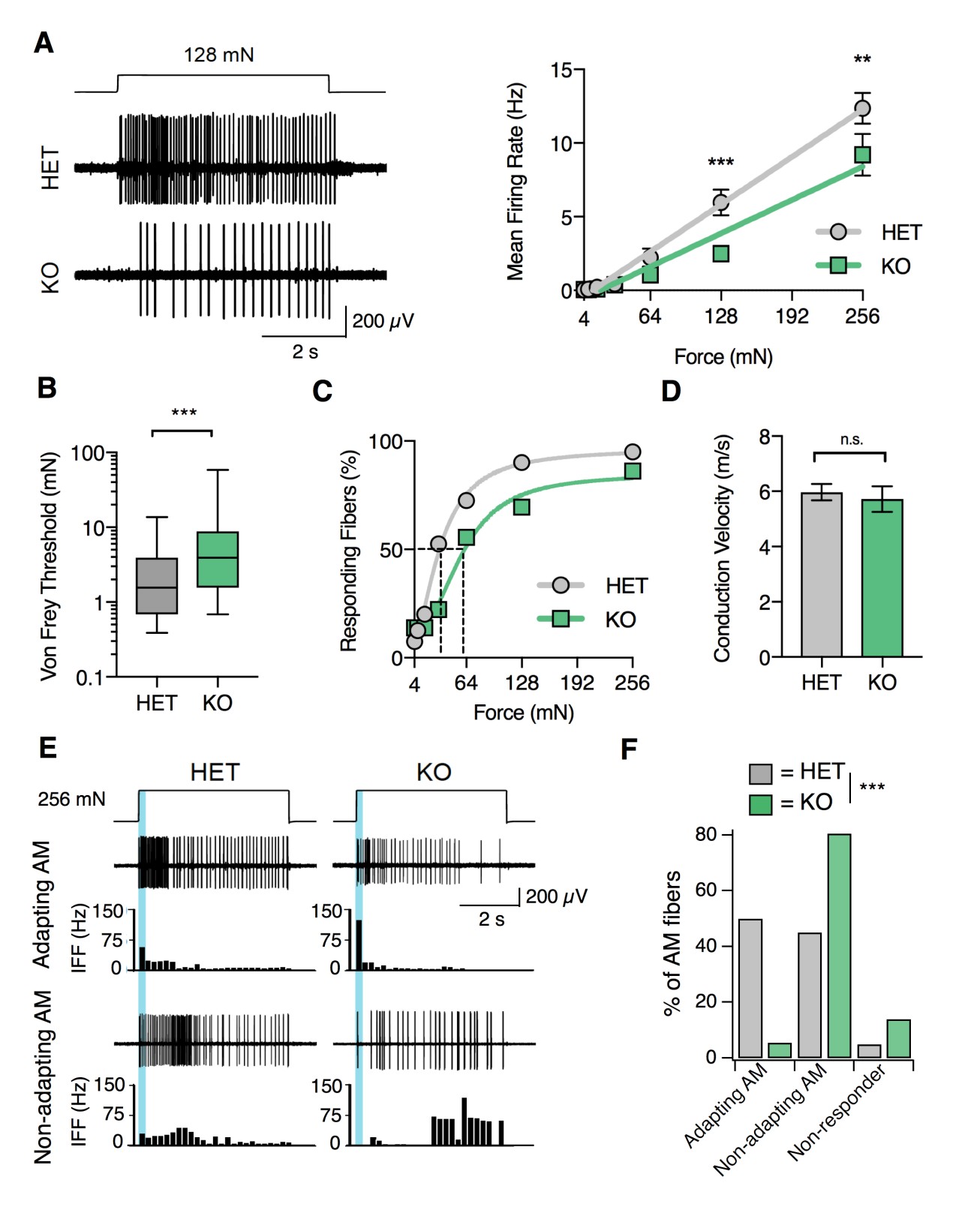

**Figure 6.** S1PR3 is required for nociceptive responses of high-threshold AM nociceptors. (**A**) (Left) Representative traces of AM fiber activity over time in ex vivo skin-saphenous nerve recording in response to stimulation (128 mN, top) from HET (middle) and KO (bottom) mice. (Right) Mean firing rate of AM fibers in response to force controlled stimulation (4, 8, 16, 32, 64, 128, 256 mN). **p=0.001, ***p=0.0002 (two-way ANOVA, Sidak's post-hoc); lines, linear regression (HET: slope = 50 Hz/N, $R^2$ = 0.99; KO: slope = 35 Hz/N, $R^2$ = 0.95). (**B**) von Frey threshold of AM fibers in S1PR3 HET and KO

*Figure 6 continued on next page*

*Figure 6 continued*

specimens. ***p<0.0001 (Mann-Whitney test); lines, median; boxes, 25–75 percentile; whiskers, min-max. (C) Cumulative response plot of AM fibers to force controlled stimulation (solid lines); four-parameter logistic fit from which half-maximal force was estimated for each genotype (dotted lines). (D) Conduction velocity (CV) of AM fibers in S1PR3 HET and KO mice. p=0.65 (two-tailed t-test); n = 40, 36 fibers; errors, mean ± SEM. (E) Representative traces and binned instantaneous firing frequencies (IFF; 200 ms bins) of Non-Adapting and Adapting AMs in response to force controlled stimulation (256 mN, top) for S1PR3 HET and KO mice; blue regions, dynamic phase of stimulation (200 ms). (F). Proportion of fibers classified by pattern of mechanically evoked responses to 256-mN stimuli: Non-Responder (HET, 2/40 fibers; KO 5/36), Non-Adapting AM (HET, 18/40; KO, 29/36), Adapting AM (HET, 20/40; KO, 2/36). Non-Responders fired action potentials to large magnitude von Frey monofilaments (<0.5 mm tip diameter), but not controlled mechanical stimulation (256 mN, 2 mm tip diameter). ***p<0.00001 (Chi-square test).
DOI: https://doi.org/10.7554/eLife.33285.013

The following source data and figure supplement are available for figure 6:

**Source data 1.** S1PR3 is required for nociceptive responses of high-threshold AM nociceptors.
DOI: https://doi.org/10.7554/eLife.33285.015
**Figure supplement 1.** S1PR3 HET AM nociceptors display normal nociceptive responses.
DOI: https://doi.org/10.7554/eLife.33285.014

mechanosensitivity of the Adapting AM population accounts for the significant reduction in force-firing relations observed at the population level in S1PR3 KO AMs (*Figure 6A*). We conclude that S1PR3 is an essential regulator of both mechanical threshold and sensitivity in a distinct population of AM nociceptors.

## S1PR3 is required for inflammatory pain hypersensitivity

Having examined the mechanisms of S1P/S1PR3 signaling in acute mechanonociception, we next sought to evaluate S1P/S1PR3 signaling in pain hypersensitivity. For this purpose, we used an experimental model of inflammatory pain triggered by Complete Freund's Adjuvant (CFA) injection into the hindpaw, which elicits infiltration of immune cells and thermal and mechanical hypersensitivity (*Ghasemlou et al., 2015*). While one previous study proposed that S1PR3 promotes injury-evoked heat and mechanical hypersensitivity, they did not measure or compare post-injury mechanical thresholds to pre-injury baselines for the knockout or control animals (*Camprubí-Robles et al., 2013*). Here, we compared development of CFA-evoked hypersensitivity between S1PR3 HET and KO littermates, since no significant behavioral differences were observed between WT and HET animals in CFA experiments ($p_{\text{von Frey}}$ = 0.12; $p_{\text{radiant heat}}$ = 0.12; two-tailed t-tests). Strikingly, S1PR3 KO mice failed to develop thermal hypersensitivity (*Figure 7A*; see *Figure 7—source data 1*) relative to heterozygous littermates at both 24 and 48 hr post-CFA injection. In stark contrast, S1PR3 KO animals developed robust mechanical hypersensitivity when thresholds were normalized to account for the dramatic baseline differences between knockouts and control animals (*Figure 7B*). Our data demonstrate that S1PR3 mediates baseline mechanical sensitivity and is not required for the development of CFA-evoked mechanical hypersensitivity.

Influx of myeloid lineage (Cd11b$^+$/Ly6G$^-$) cells is required for the development of mechanical hypersensitivity in the CFA model (*Ghasemlou et al., 2015*). In the immune system, S1P signaling via S1PR1 plays a key role in immune cell migration (*Matloubian et al., 2004*). Consistent with the development of mechanical hypersensitivity, flow cytometry experiments showed robust infiltration of immune cells into hindpaw skin from both S1PR3 KO and littermate controls (*Figure 7—figure supplement 1A–B*). These data suggest that the phenotypes observed in S1PR3 KO mice cannot be attributed to compromised immune cell infiltration.

Next, we tested whether active S1P/S1PR3 signaling was required to maintain CFA-evoked thermal hypersensitivity using pharmacology. Acute blockade of S1P production with SKI II or S1PR3 with TY also reversed CFA heat hypersensitivity (*Figure 7C*), demonstrating that peripheral S1P actively signals via S1PR3 to promote CFA-evoked heat hypersensitivity. Furthermore, acute S1P/S1PR3 blockade with SKI II or TY elevated mechanical thresholds to pre-CFA, baseline levels (*Figure 7D*) showing that S1PR3 tunes mechanical pain under normal and inflammatory conditions. These results are consistent with the distinct roles for AM and C nociceptors in mechanical pain. Under normal conditions, AM nociceptors set mechanical pain thresholds (*Osteen et al., 2016*; *Abrahamsen et al., 2008*). By contrast, under inflammatory conditions, the combined activity of both non-sensitized AM fibers and sensitized C fibers determines overall post-inflammatory

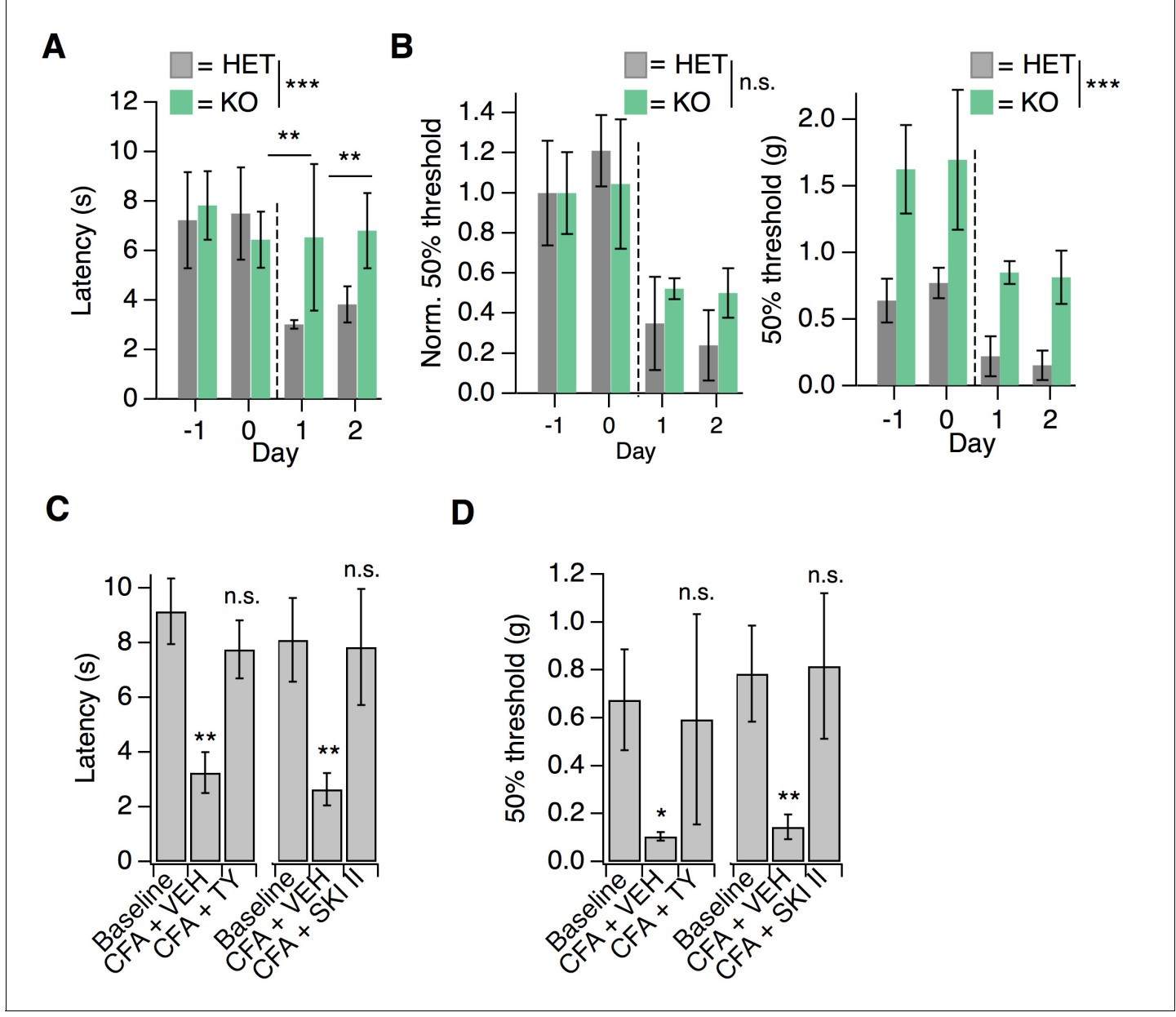

**Figure 7.** S1PR3 is dispensable for development of chronic mechanical hypersensitivity. (**A**) Thermal latency before and after CFA treatment (indicated by dotted line); $p_{genotype}$ = 0.0053 (two-way ANOVA; N = 5 mice per genotype). Sidak's multiple comparison between genotypes for specific time points indicated on graph. Error bars represent mean ± SD. (**B**) (Left) Normalized 50% withdrawal threshold before and after CFA treatment (indicated by dotted line); p(genotype)<0.001 (two-way ANOVA). (Right) 50% withdrawal thresholds for same experiment (p(genotype)=0.1634; two-way ANOVA). (**C**) (Left) Thermal latency assessed before ('Baseline') and 24 hr post CFA injection with either vehicle (CFA + VEH) or TY 52156 (CFA + TY) acutely administered; p<0.0001 (one-way ANOVA N = 5 mice per treatment). (Right) Thermal latency assessed before and after CFA injection with either vehicle (CFA + VEH) or SKI II (CFA + SKI II) acutely administered on Day 1; p<0.0001 (one-way ANOVA; N = 5–7 mice per treatment). Dunnett's test comparisons to baseline are indicated on graph. Error bars represent mean ± SD. (**D**) (Left) 50% withdrawal threshold assessed before and 24 hr post CFA injection with either vehicle (CFA + VEH) or TY 52156 (CFA + TY) acutely administered on Day 1; p<0.0001 (one-way ANOVA; N = 5 mice per treatment). Dunnett's test comparisons to baseline are indicated on graph. (Right) 50% withdrawal threshold assessed before and 24 hr post CFA injection with either vehicle (CFA + VEH) or SKI II (CFA + SKI II) acutely administered; p-values indicated on graph (two-tailed unpaired t-test; N = 5 mice per group.

DOI: https://doi.org/10.7554/eLife.33285.016

The following source data and figure supplement are available for figure 7:

**Source data 1.** S1PR3 is dispensable for development of chronic mechanical hypersensitivity.

DOI: https://doi.org/10.7554/eLife.33285.018

*Figure 7 continued on next page*

*Figure 7 continued*

**Figure supplement 1.** S1PR3 KO animals display normal CFA-evoked immune cell recruitment.

DOI: https://doi.org/10.7554/eLife.33285.017

mechanical thresholds (*Lennertz et al., 2012*; *Abrahamsen et al., 2008*). Consistent with this model, acute blockade of S1P production or S1PR3 activity under normal conditions induces mechanical hyposensitivity and under inflammatory conditions returns mechanical sensitivity to normal levels.

## Discussion

We now show that S1P signaling via S1PR3 is a key pathway that tunes mechanical pain sensitivity. Overall, our data reveal two new key findings. First, S1P/S1PR3 sets baseline mechanical pain thresholds. Depletion of baseline, endogenous S1P induces mechanical hyposensitivity and nanomolar levels of exogenous S1P are sufficient to restore normal mechanical pain sensitivity after depletion. Second, elevated micromolar S1P levels, such as those produced during inflammation or disease, promote thermal, but not mechanical hypersensitivity. The effects of S1P on acute mechanical pain and thermal hypersensitivity are completely lost in S1PR3 knockout animals, which are otherwise normal with respect to other somatosensory behaviors.

What is the local source of S1P in the skin that constitutively modulates mechanical pain? Even in the mature field of S1P signaling in the vascular and immune systems, the cellular source of S1P, while an intriguing question, remains unclear. All cells in the body, including somatosensory neurons, immune cells, and skin cells, express sphingosine kinases 1 and 2 which are essential for S1P production (*Chalfant and Spiegel, 2005*). Deletion of both kinases is lethal and attempts to conditionally knockout these kinases fail to completely eliminate S1P in tissues (*Pappu et al., 2007*). While RNA seq data suggests that somatosensory neurons contain all of the enzymatic machinery required to produce and export local S1P (*Usoskin et al., 2015*; *Morita et al., 2015*), future work will be needed to identify the key cell types that are important for maintaining baseline S1P levels in the skin to regulate mechanical sensitivity and for increasing S1P under inflammatory/injury conditions to promote pain hypersensitivity.

Recent studies have identified distinct populations of AM nociceptors that are required for mechanical pain (*Arcourt et al., 2017*; *Ghitani et al., 2017*). Likewise, it was discovered that a subset of somatostatin-expressing spinal interneurons is required for mechanical pain transduction (*Duan et al., 2014*). Although these papers delineate the cells and circuitry of mechanical pain, the molecular underpinnings of mechanonociception in the periphery are poorly understood. While the identity of the transduction channel(s) in AM nociceptors remains enigmatic, understanding molecular mechanisms that regulate excitability will no doubt provide key insights into the function and specialization of the diverse subtypes of mechanosensitive nerve fibers. For example, although *Piezo2*-hypomorphic animals exhibit normal mechanical pain behaviors (*Ranade et al., 2014*), ex vivo skin-nerve recordings show that their AM nociceptors display decreased force-responsiveness (*Ranade et al., 2014*), and a recent study found that subpopulations of sensory neurons express different splice variants of *Piezo2* that exhibit different force sensitivities (*Szczot et al., 2017*). These studies suggest that mechanosensitive neurons exhibit functional specialization on multiple levels. Our study demonstrates that S1PR3 is indispensable for normal function of AM nociceptors, including the adapting AM population, recently discussed in Arcourt et al., that innervates the epidermis and encodes noxious touch (*Arcourt et al., 2017*). We show that S1PR3 signaling modulates KCNQ2/3 channels to regulate excitability of these A mechanonociceptors (*Figure 8*).

GPCR-mediated inhibition of KCNQ2/3 potassium channels is a well-known mechanism by which neuronal excitability is regulated (*Passmore et al., 2003*). Other studies have shown that KCNQ channels mediate excitability of Aδ fibers (*Schütze et al., 2016*; *Passmore et al., 2012*) and are required for normal mechanonociceptive responses in dorsal horn neurons receiving Aδ input (*Passmore et al., 2012*), and that opening KCNQ2/3 channels directly with retigabine alleviates pain in vivo (*Xu et al., 2010*; *Hayashi et al., 2014*; *Blackburn-Munro and Jensen, 2003*). Our results not only complement previous work implicating KCNQ2/3 channels in pain, but also define the upstream mechanisms that promote the regulation of KCNQ2/3 channels to tune mechanical pain thresholds.

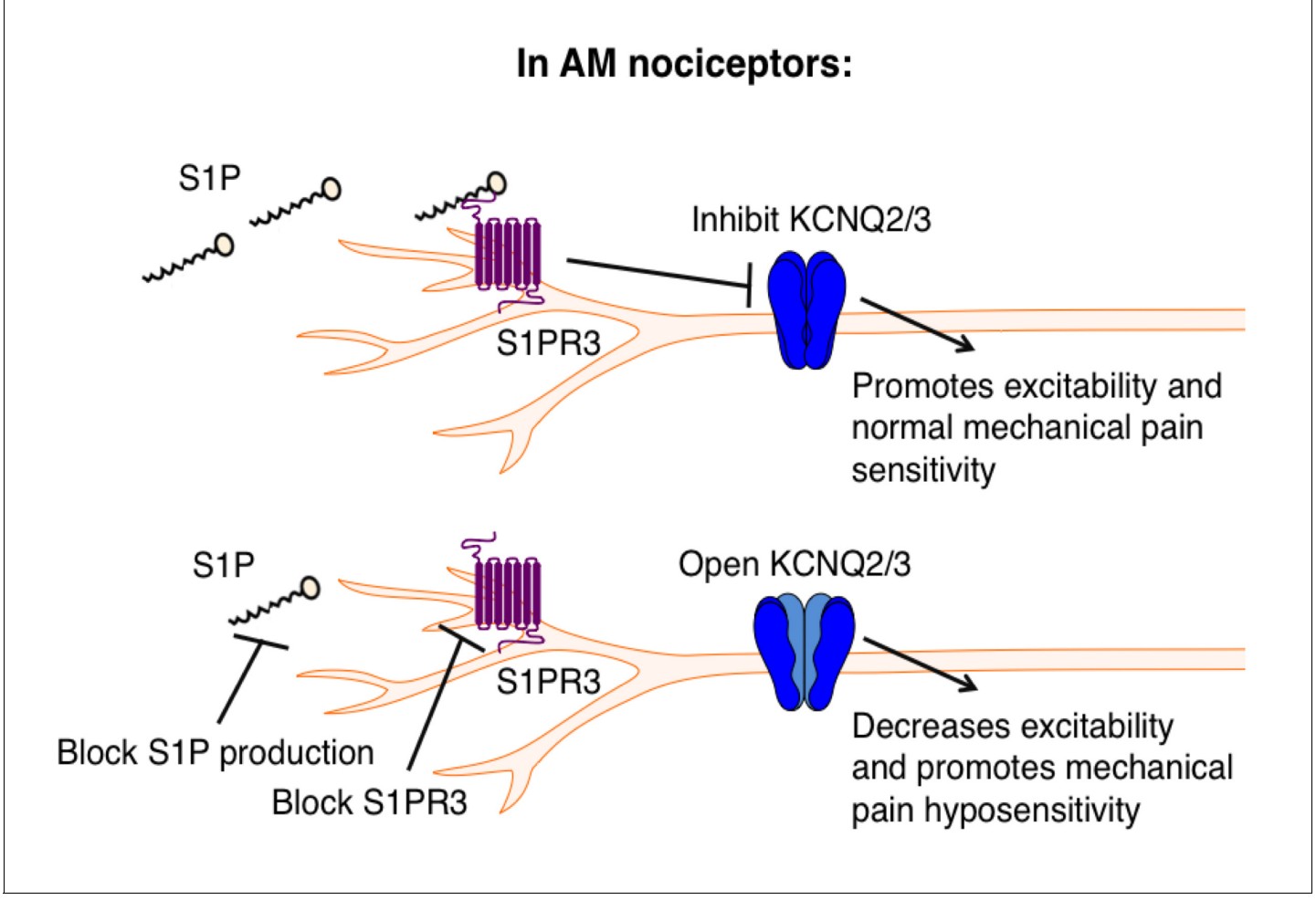

**Figure 8.** Proposed model illustrating a key role for S1PR3 in regulating mechanical pain in AM nociceptors. (Top) S1P promotes activation of S1PR3, which leads to inhibition of KCNQ2/3 currents and promotes normal mechanical pain sensitivity. (Bottom) Diminished S1P or S1PR3 antagonism alleviates inhibition of KCNQ2/3, leading to mechanical pain hyposensitivity.

DOI: https://doi.org/10.7554/eLife.33285.019

Our data thus highlight S1PR3 as a novel and attractive target for the treatment of mechanical pain and describe a new signaling pathway regulating AM nociceptor excitability.

Interestingly, the neurons that innervate the ultra-sensitive tactile organ of the star-nosed mole are highly enriched in transcripts for S1PR3 and KCNQ channels, as well as for a variety of other potassium channels (*Gerhold et al., 2013*). While it is difficult to directly examine the physiological basis for heightened mechanosensitivity in the star-nosed mole, S1PR3-dependent modulation of KCNQ may represent an important mechanism underlying the high tactile sensitivity of the star organ. Moreover, the link between *S1pr3* and *Kcnq2/3* is echoed in single-cell RNA seq datasets from mouse DRG neurons, which show co-expression of *S1pr3* and *Kcnq2/3* in a subset of myelinated mechanoreceptors (*Usoskin et al., 2015*). These cells are distinct from the *S1pr3/Trpv1* subset that mediates S1P-evoked acute pain and heat hypersensitivity. In addition to being transcriptionally distinct, we show that the mechanisms underlying S1P's activities in these cells are functionally distinct. Finally, *S1pr3* and *Kcnq2/3* are highly expressed in human sensory ganglia (*Ray et al., 2018*; *Flegel et al., 2015*) and recordings from human stem cell-derived sensory neurons show that KCNQ2/3 channels play a key role in mediating their excitability (*Young et al., 2014*). Thus S1PR3 signaling may represent a new target for modulating mechanical pain.

Previous studies of S1P signaling in DRG neurons focused on S1P-evoked excitation of small diameter and/or capsaicin-sensitive neurons, and pain behaviors triggered by elevated S1P. While our new

data affirms the effects S1P in thermal nociceptors observed by others, our manuscript highlights a novel effect of baseline levels of S1P in modulation of rheobase and KCNQ2/3 currents in mechanonociceptors. We are also the first to examine the role of S1PR3 in a variety of somatosensory behaviors under normal conditions (non-injury, no algogen injection), and to demonstrate a key role for S1PR3 in mechanonociception. We also go beyond previous studies (*Mair et al., 2011*; *Camprubí-Robles et al., 2013*; *Finley et al., 2013*; *Weth et al., 2015*) in showing that elevated S1P selectively promotes thermal and not mechanical hypersensitivity. Others have shown that there are distinct cells and molecular pathways that trigger thermal versus mechanical hypersensitivity. For example, thermal hypersensitivity in the CFA model is dependent on TRPV1 ion channels (*Caterina et al., 2000*) and independent of immune cell infiltration (*Ghasemlou et al., 2015*). By contrast, infiltration of a subset of myeloid immune cells is required for mechanical hypersensitivity in an inflammatory pain model (*Ghasemlou et al., 2015*). Here we show that S1P via S1PR3 signaling is a key component of the inflammatory soup that triggers thermal hypersensitivity in the CFA model. Our observation that S1PR3 KO animals display normal immune cell infiltration and develop mechanical hypersensitivity after CFA is consistent with these previous studies showing distinct mechanisms of inflammatory thermal and mechanical hypersensitivity.

Outside of the nervous system, S1P signaling via S1PR1 allows for the continuous circulation of lymphocytes between blood, lymph, and peripheral tissues (*Matloubian et al., 2004*). Our findings that S1P plays a key role in noxious mechanosensation are in line with recent studies showing that sensory neurons co-opt classical immune pathways to drive chronic itch or pain (*Oetjen et al., 2017*; *Pinho-Ribeiro et al., 2017*). What distinguishes this study from the others is that S1P signaling is critical for acute mechanical pain, even in the absence of inflammation or exogenously elevated S1P. In the immune system, disruptions in S1P levels or S1PR1 signaling result in significant immune dysfunction and disease (*Donoviel et al., 2015*; *Gräler and Goetzl, 2004*; *Olivera et al., 2013*). Accordingly, in the somatosensory system, excessive, high levels of S1P (micromolar), such as those present in inflammation, evokes thermal pain and sensitization. Intermediate, baseline levels (nanomolar) regulate AM excitability and are required for normal mechanical pain sensation. By contrast, lowering S1P levels reduces mechanical pain sensation, while sparing innocuous touch sensation.

We propose that S1PR3 signaling may contribute to a variety of inflammatory diseases. S1P has been linked to a wide range of human inflammatory disorders (*Allende et al., 2011*; *Kunkel et al., 2013*; *Roviezzo et al., 2015*; *Liang et al., 2013*; *Rivera et al., 2008*; *Myśliwiec et al., 2017*; *Checa et al., 2015*). Canonically, S1P signaling via S1PR1 is thought to promote inflammation via the immune system (*Kunkel et al., 2013*), however, we propose that S1P signaling via S1PR3 in neurons may also contribute to inflammatory disease. Indeed, fingolimod, a non-selective S1PR modulator, is prescribed as an immunosuppressant for multiple sclerosis treatment (*Brinkmann et al., 2010*), but the possibility that some of its therapeutic effects may also be mediated via the nervous system has not been fully explored. Likewise, one study found that intrathecal fingolimod reduces bone cancer pain (*Grenald et al., 2017*), and while analgesia was attributed to effects on S1PR1 in glia, some of the benefits may be due to S1PR3 signaling in DRG neurons. S1PR3 antagonism may also be useful in the treatment of inflammatory pain due to its selective dampening of acute mechanical pain and inflammatory thermal hypersensitivity, while preserving innocuous touch and normal thermal sensitivity. S1PR3 inhibitors may also be beneficial for treating other inflammatory disorders where S1PR3-expressing somatosensory neurons have been shown to contribute to neurogenic inflammation, such as asthma (*Tränkner et al., 2014*). Our study demonstrates a crucial role for S1P signaling in the peripheral nervous system and highlights the potential of S1PR3 as a target for future pain therapies.

## Materials and methods

### Key resources table

| Reagent type (species) or resource | Designation | Source or reference | Identifiers | Additional information |
|---|---|---|---|---|
| strain, strain background (C57BL/6J) | C57BL/6J; WT; wild-type | The Jackson Laboratory | Jackson Stock #: 000664; RRID:IMSR_JAX:000664 | |

*Continued on next page*

*Continued*

| Reagent type (species) or resource | Designation | Source or reference | Identifiers | Additional information |
|---|---|---|---|---|
| strain, strain background (B6.129S6-S1pr3tm1Rlp/Mmnc) | S1PR3 KO; S1pr3-/- | MMRRC Repository; https://www.ncbi.nlm.nih.gov/pubmed/15138255; PMID: 15138255 | B6.129S6-S1pr3tm1Rlp/Mmnc; MMRRC Stock #: 012038-UNC; RRID: MMRRC_012038-UNC | |
| strain, strain background (B6.Cg-S1pr3tm1.1Hrose/J) | *S1pr3-mCherry; S1pr3mcherry/+* | The Jackson Laboratory | B6.Cg-S1pr3tm1.1Hrose/J; Jackson Stock #: 028624; RRID:IMSR_JAX:028624 | |
| antibody (Living Colors DsRed Rabbit Polyclonal Antibody) | Rabbit anti-DsRed | Clontech | RRID:AB_10013483; Cat # 632496 | |
| antibody (Chicken polyclonal to Neurofilament heavy polypeptide) | Chicken anti-NefH | Abcam | RRID:AB_304560; Cat # ab4680 | |
| antibody (Chicken polyclonal to beta III Tubulin) | Chicken anti-β-tubulin III | Abcam | RRID:AB_10899689; Cat # ab107216 | |
| antibody (Mouse monoclonal [13C4/I3 C4] to PGP9.5) | Mouse anti-PGP9.5 | Abcam | RRID:AB_306343; Cat # ab8189 | |
| antibody (Rabbit polyclonal to EDG3) | Rabbit anti-S1PR3 | Abcam | RRID:AB_732070; Cat # ab38324 | |
| antibody (Mouse monoclonal to NF200) | Mouse anti-NF200 | Sigma-Aldrich | RRID:AB_260781; Cat # N5389 | |
| antibody (Chicken polyclonal to Peripherin) | Chicken anti-Peripherin | Abcam | RRID:AB_777207; Cat # ab39374 | |
| antibody (Goat Anti-Mouse IgG H and L Alexa Fluor 488) | Goat anti-Mouse Alexa 488 | Abcam | RRID:AB_2688012; Cat # ab150117 | |
| antibody (Goat anti-Chicken IgY (H + L) Secondary Antibody, Alexa Fluor 488) | Goat anti-Chicken Alexa 488 | ThermoFisher Scientific | RRID:AB_2534096; Cat # A-11039 | |
| antibody (Goat anti-Rabbit IgG (H + L) Secondary Antibody, Alexa Fluor 594) | Goat anti-Rabbit Alexa 594 | Invitrogen | RRID:AB_2556545; Cat # R37117 | |
| sequence-based reagent | *S1pr3* Type I Probe | ThermoFisher Scientific; Affymetrix | Assay ID: VB1-19668-VC | |
| sequence-based reagent | *Scn1a* Type 6 Probe | ThermoFisher Scientific; Affymetrix | Assay ID: VB6-18173-VC | |
| sequence-based reagent | *Npy2r* Type 6 Probe | ThermoFisher Scientific; Affymetrix | Assay ID: VB6-3197254-VC | |
| sequence-based reagent | *Piezo2* Type 6 Probe | ThermoFisher Scientific; Affymetrix | Assay ID: VB6-18046-VC | |
| sequence-based reagent | *Trpv1* Type 6 Probe | ThermoFisher Scientific; Affymetrix | Assay ID: VB6-18246-VC | |
| sequence-based reagent | *Trpa1* Type 6 Probe | ThermoFisher Scientific; Affymetrix | Assay ID: VB6-16610-VC | |
| commercial assay or kit (ViewRNA ISH Tissue Assay Kit (2-plex)) | ViewRNA ISH Tissue Assay Kit | ThermoFisher Scientific; Affymetrix | Cat # QVT0012 | |
| chemical compound, drug (Sphingosine-1-phosphate) | Sphingosine 1-phosphate; S1P | Tocris Bioscience; Avanti Polar Lipids | CAS 26993-30-6; Cat # 1370; Cat # 860641 | |
| chemical compound, drug (TY 521256) | TY 52156 | Tocris Bioscience | CAS 934369-14-9; Cat # 5328 | |
| chemical compound, drug (SKI II) | SKI II | Tocris Bioscience | CAS 312636-16-1; Cat # 2097 | |
| chemical compound, drug (Histamine dihydrochloride) | Histamine | Sigma-Aldrich | CAS 56-92-8; Cat # H7250 | |

*Continued on next page*

*Continued*

| Reagent type (species) or resource | Designation | Source or reference | Identifiers | Additional information |
|---|---|---|---|---|
| chemical compound, drug (Chloroquine diphosphate) | Chloroquine | Sigma-Aldrich | CAS 50-63-5; Cat # C6628 | |
| chemical compound, drug (E-Capsaicin) | Capsaicin | Tocris Bioscience | CAS 404-86-4; Cat # 0462 | |
| chemical compound, drug (Dimethyl sulfoxide) | DMSO | Sigma-Aldrich | Cat # 8418–100 mL | |
| chemical compound, drug (Methanol) | Methanol | Sigma-Aldrich | CAS 67-56-1; Cat # 34860 | |
| chemical compound, drug (Linopirdine dihydrochloride) | Linopirdine | Tocris Bioscience | CAS 113168-57-3; Cat # 1999 | |
| chemical compound, drug (XE 991 dihydrochloride) | XE 991 | Tocris Bioscience | CAS 122955-13-9; Cat # 2000 | |
| chemical compound, drug (W146) | W146 | Tocris Bioscience | CAS 909725-61-7; Cat # 3602 | |
| chemical compound, drug (Freund's Adjuvant, Complete) | Complete Freund's Adjuvant; CFA | Sigma-Aldrich | Cat # F5881 | |
| chemical compound, drug (Formaldehyde, 16%, methanol free, Ultra Pure) | Paraformaldehyde; PFA | Polysciences, Inc. | Cat # 18814–10 | |
| chemical compound, drug (Tissue Tek Optimal cutting temperature compound (OCT)) | OCT | Sakura Finetek USA | Cat # 4583 | |
| chemical compound, drug (Triton X-100 solution) | Triton X-100 | BioUltra | CAS 9002-93-1; Cat # 93443 | |
| chemical compound, drug (Phosphate-buffered saline (PBS), pH 7.4) | PBS | Gibco | Cat # 10010023 | |
| chemical compound, drug (Benzyl benzoate) | Benzyl benzoate | Sigma-Aldrich | CAS 120-51-4; Cat # B6630 | |
| chemical compound, drug (Benzyl alcohol) | Benzyl alcohol | Sigma-Aldrich | CAS 100-51-6; Cat # 305197 | |
| chemical compound, drug (Sucrose) | Sucrose | Sigma-Aldrich | CAS 57-50-1; Cat # S0389 | |
| chemical compound, drug (LIVE/DEAD Fixable Aqua Dead Cell Stain Kit, for 405 nm excitation) | Aqua | ThermoFisher Scientific | Cat # L34957 | |
| chemical compound, drug (Isoflurane, USP) | Isoflurane | Piramal | CAS 26675-46-7 | |
| chemical compound, drug (4',6-Diamidino-2-Phenylindole, Dihydrochloride) | DAPI | ThermoFisher Scientific | CAS 28718-90-3; Cat # 1306 | |
| chemical compound, drug (Fluoromount-G, with DAPI) | Fluoromount-G, with DAPI | ThermoFisher Scientific | Cat # 00-4959-52 | |
| antibody (CD117 (c-Kit) Monoclonal Antibody (2B8), Biotin) | c-Kit-Biotin | eBioscience | RRID:AB_466569; Cat # 13-1171-82 | |
| antibody (FceR1 alpha Monoclonal Antibody (AER-37 (CRA1)), PE, eBioscience) | FceRI-PE | eBioscience | RRID:AB_10804885; Cat # 12-5899-42 | |
| antibody (CD49b (Integrin alpha 2) Monoclonal Antibody (DX5), PE-Cyanine7, eBioscience) | CD49b-PECy7 | eBioscience | RRID:AB_469667; Cat # 25-5971-82 | |

*Continued on next page*

*Continued*

| Reagent type (species) or resource | Designation | Source or reference | Identifiers | Additional information |
|---|---|---|---|---|
| antibody (Anti-Siglec-F-APC, mouse (clone: REA798)) | SiglecF-APC | Miltenyi Biotech | RRID:AB_2653441; Cat # 130-112-333 | |
| antibody (Streptavidin FITC) | SA-FITC | eBioscience | RRID:AB_11431787; Cat # 11-4317-87 | |
| antibody (Ly-6C Monoclonal Antibody (HK1.4), PerCP-Cyanine5.5, eBioscience) | Ly6C-PerCP | eBioscience | RRID:AB_1518762; Cat # 45-5932-82 | |
| antibody (Pacific Blue anti-mouse/human CD11b Antibody) | CD11b-PB | BioLegend | RRID:AB_755985; Cat # 101223 | |
| antibody (Brilliant Violet 785 anti-mouse Ly-6G Antibody) | Ly6G-BV785 | BioLegend | RRID:AB_2566317; Cat # 127645 | |
| antibody (CD45.2 Monoclonal Antibody (104), Alexa Fluor 700, eBioscience) | CD45.2-AF700 | eBioscience | RRID:AB_657752; Cat # 56-0454-82 | |
| software, algorithm (Igor Pro version 6.3) | IgorPro | WaveMetrics | https://www.wavemetrics.com/order/order_igordownloads6.htm | |
| software, algorithm (Microsoft Excel 2011) | Microsoft Excel | Microsoft | https://www.microsoft.com/en-us/store/d/excel-2016-for-mac/ | |
| software, algorithm (pClamp 10) | pClamp | Axon | http://mdc.custhelp.com/app/answers/detail/a_id/18779/~/axon%E2%84%A2-pclamp%E2%84%A2-10-electrophysiology-data-acquisition-%26-analysis-software | |
| software, algorithm (MetaFluor 7.8) | MetaFluor | Molecular Devices | https://www.moleculardevices.com/systems/metamorph-research-imaging/metafluor-fluorescence-ratio-imaging-software | |
| software, algorithm (MATLAB) | MATLAB | MathWorks | https://www.mathworks.com/downloads/ | |
| software, algorithm (FIJI) | FIJI | NIH | https://imagej.net/Fiji/Downloads | |
| software, algorithm (LabChart Software) | LabChart Software | AD Instruments | https://www.adinstruments.com/products/labchart | |
| software, algorithm (Graphpad Prism 7) | Graphpad Prism | Graphpad | https://www.graphpad.com/scientific-software/prism/ | |
| software, algorithm (FlowJo 10.4.2) | FlowJo | FlowJo | https://www.flowjo.com/solutions/flowjo/downloads | |
| software, algorithm (custom) | custom-made software in MATLAB | this paper | NA | https://github.com/buh2003/SpikeSortingPCA_DBSCAN (*Hoffman, 2018*; copy archived at https://github.com/elifesciences-publications/SpikeSortingPCA_DBSCAN) |
| other (Bovine serum albumin, cold ethanol fraction, pH 5.2,≥96%) | BSA | Sigma-Aldrich | CAS 9048-46-8; Cat # A4503 | |
| other (Isolectin B4 (Bandeireia simplicifolia), FITC-conjugate) | IB4-FITC; IB4 | Enzo Life Sciences | Cat # ALX-650–001F-MC05 | |
| other (Normal Goat Serum) | NGS | Abcam | Cat # ab7481 | |

*Continued on next page*

*Continued*

| Reagent type (species) or resource | Designation | Source or reference | Identifiers | Additional information |
|---|---|---|---|---|
| peptide, recombinant protein (δ-theraphototoxin-Hm1a) | Hm1a | other; https://www.ncbi.nlm.nih.gov/pmc/articles/PMC4919188/; PMID: 4919188 | NA | obtained from the laboratory of David Julius (UCSF) |

## Behavioral studies and mice

$S1pr3^{mcherry/+}$ and $S1pr3^{-/-}$ mice were obtained from Jackson Laboratory and backcrossed to C57bl6/J. Wherever possible, wild-type/heterozygous (*S1pr3*) littermate controls were used in behavioral experiments. Mice (20–25 g) were housed in 12 hr light-dark cycle at 21°C. Mice were singly housed one week prior to all behavioral experiments and were between 8–10 weeks at the time of the experiment. All mice were acclimated in behavioral chambers (IITC Life Sciences) on two subsequent days for 1 hr prior to all behavioral experiments.

Itch and acute pain behavioral measurements were performed as previously described (*Morita et al., 2015*; *Wilson et al., 2013*; *Tsunozaki et al., 2013*). Mice were shaved one week prior to itch behavior. Compounds injected: 500 µM TY 52156 (Tocris), 50 µM SKI II (Tocris), 0.2–10 µM S1P (Tocris, Avanti Polar Lipids), 50 mM chloroquine (Sigma), and 27 mM histamine (Tocris) in PBS with either 0.01–0.1% Methanol- (S1P) or 0.1–0.5% DMSO-PBS (all other compounds) vehicle controls. Pruritogens were injected using the cheek model (20 µL) of itch, as previously described (*Shimada and LaMotte, 2008*). Behavioral scoring was performed while blind to experimental condition and mouse genotype. All scratching and wiping behavior videos were recorded for 1 hr. Itch behavior was scored for the first 30 min and acute pain was scored for the first five minutes. Bout number and length were recorded.

For radiant heat and von Frey hypersensitivity behavior, drugs were injected intradermally into the plantar surface of the hindpaw (20 µL). Radiant heat assays were performed using the IITC Life Science Hargreaves test system. Mechanical threshold was measured using calibrated von Frey monofilaments (Touch Test) on a metal grate platform (IITC). Von Frey was performed as previously described (*Tsunozaki et al., 2013*; *Chaplan et al., 1994*) using the up-down method (*Dixon, 1965*) while blinded to compound injected and genotype, or a descending force-series of 4 trials per force from 0.4 g to 6 g. Valid responses for both von Frey and radiant heat included fast paw withdrawal, licking/biting/shaking of the affected paw, or flinching. For radiant heat and von Frey, mice were allowed to acclimate on platform for 1 hr before injection. Measurements were taken 15 min pre-injection and 20–30 min post-injection for all compounds used.

The pinprick assay (*Duan et al., 2014*) was conducted on a von Frey testing platform (IITC). The mouse hindpaw was poked with a 31 g syringe needle without breaking the skin to induce fast acute mechanical pain. Each paw was stimulated 10 times with the needle, with five minutes rest in between trials, and the % withdrawal (fast withdrawal, licking/biting/shaking of paw, squeaking, and/or flinching) was calculated from the total number of trials.

The tape assay was conducted according to previously described methods (*Ranade et al., 2014*). Number of attempts to remove a 3 cm piece of lab tape was recorded for 10 min after manual tape application to the rostral back. Scorer and experimenter were blinded to genotype.

For righting reflex measurements, age-matched $S1pr3^{-/-}$ and $^{+/+}$ P6-7 neonates were used. Briefly, pups were overturned one at a time on the home cage lid while experimenter was blinded to genotype. The time to righting was measured to the nearest $1/10^{th}$ of a second with a stopwatch.

For the CFA model of hypersensitivity, mice were lightly anesthetized with isoflurane (2%) and injected with 15 µL CFA (Sigma) into one hindpaw using a Hamilton syringe (30 g) at 5pm. Radiant heat latencies and von Frey 50% withdrawal thresholds were recorded one day prior to CFA, the morning of CFA (prior to injection), and one and two days post-CFA. Von Frey measurements were acquired before radiant heat latencies, and mice were allowed a one-hour recovery period in home cage with access to food and water in between testing. Both ipsilateral and contralateral paw were measured. Experimenter was blind to genotype for injections and recording.

All behavior experiments were carried out using age-matched or littermate cohorts of male mice and conducted between 8 am and 1 pm. Mice were tested in 4-part behavior chambers (IITC Life Sciences) with opaque dividers (TAP Plastics) with the exception of righting reflex measurements.

Scratching and wiping behaviors were filmed from below using high-definition cameras. All experiments were performed under the policies and recommendations of the International Association for the Study of Pain and approved by the University of California, Berkeley Animal Care and Use Committee.

### In situ hybridization (ISH)

Fresh DRG were dissected from 8 to 12 week old mice, flash frozen in OCT embedding medium, and sectioned at 14 µm onto slides. ISH was performed using Affymetrix Quantigene ViewISH Tissue 2-plex kit according to manufacturer's instructions with Type 1 (*S1pr3*) and Type 6 (all other) probes. The following probes against mouse mRNAs were created by Affymetrix and used for ISH: *S1pr3, Scn1a, Npy2r, Piezo2, Trpv1, Trpa1.*

### Immunohistochemistry (IHC) of DRG

DRG were dissected from 8 to 12 week old adult mice and post-fixed in 4% PFA for one hour. DRG were cryo-protected overnight at 4°C in 30% sucrose-PBS, embedded in OCT, and then cryosectioned at 12 µm onto slides. Briefly, slides were washed 3x in PBST (0.3% Triton X-100), blocked in 2.5% horse serum +2.5% BSA PBST, and incubated overnight at 4°C in 1:1000 primary antibody in PBST +0.5% horse serum +0.5% BSA. Slides were washed 3X in PBS then incubated 1–2 hr at RT in 1:1000 secondary antibody. Slides were washed 3X in PBS and mounted in Fluoromount-G +DAPI with No. 1.5 coverglass. Primary antibodies used: Rabbit anti-DsRed (Clontech #632496), Rabbit anti-S1PR3 (Abcam #38324; #108370), Mouse anti-NF200 (Sigma #N5389), Chicken anti-Peripherin (Abcam #39374). Secondary antibodies used: Goat anti-Mouse Alexa 488 (Abcam #150117), Goat anti-Chicken Alexa 488 (ThermoFisher #A11039), Goat anti-Rabbit Alexa 594 (Invitrogen #R37117). Isolectin B4 (IB4)-FITC (Enzo Life Sciences #ALX-650–001F-MC05) was also used. Slides were mounted in Fluoromount with No. 1.5 coverglass. Imaging of DRG ISH and IHC experiments, and all live-cell imaging, was performed on an Olympus IX71 microscope with a Lambda LS-xl light source (Sutter Instruments). For DRG ISH and IHC analysis, images were analyzed using FIJI software. Briefly, DAPI-positive cells were circled and their fluorescence intensity (AFU) for all channels was plotted against cell size using Microsoft Excel software. Co-labeling analysis was performed using FIJI. Intensity thresholds were set based on the negative control (no probe) slide. Cells were defined as co-expressing if their maximum intensities exceeded the threshold for both channels of interest.

### IHC of sectioned skin

Skin was dissected from 8 week old adult mice and post-fixed in 4% PFA for 30 min at RT. DRG were cryo-protected overnight at 4°C in 30% sucrose-PBS, embedded in OCT, and then sectioned at 18 µm onto slides. Briefly, slides were blocked in 5% normal goat serum in PBST (0.1% Triton X-100) and incubated overnight at 4°C in 1:1000 primary antibody in blocking buffer. Slides were washed 3X in PBS then incubated 45 min at RT in 1:1000 secondary antibody. Slides were washed 5X in PBS and mounted in Fluoromount-G +DAPI with No. 1.5 coverglass. Primary antibodies used: Rabbit anti-DsRed (Clontech #632496), Chicken anti-NefH (Abcam #4680), Chicken anti-β-tubulin III (Abcam #107216), mouse anti-PGP9.5 (Abcam #8189). Secondary antibodies used: Goat anti-Mouse Alexa 488 (Abcam #150117), Goat anti-Rabbit Alexa 594 (Invitrogen #R37117), Goat anti-Chicken Alexa 488 (ThermoFisher #A11039). For co-localization analysis, only fibers for which >50% of the length of the visible fiber contained co-localized (white) pixels were counted. Image analysis was performed using FIJI.

### Whole mount skin IHC

Staining was performed according to *Marshall et al. (2016)*. Briefly, 8-week-old mice were euthanized and the back skin was shaved, depilated, and tape-stripped. The removed skin was fixed overnight in 4% PFA, then washed in PBS (3X for 10 min each). Dermal fat was scraped away with a scalpel and skin was washed in PBST (0.3% Triton X-100; 3X for two hours each) then incubated in 1:500 primary antibody (Rabbit anti DsRed: Clontech #632496; Chicken anti-Nefh: Abcam #4680) in blocking buffer (PBST with 5% goat serum and 20% DMSO) for 5.5 days at 4°C. Skin was washed as before and incubated in 1:500 secondary antibody (Goat anti-Rabbit Alexa 594; Invitrogen #R37117; Goat anti-Chicken Alexa 488; ThermoFisher #A11039) in blocking buffer for 3 days at 4°C. Skin was

washed in PBST, serially dried in methanol: PBS solutions, incubated overnight in 100% methanol, and finally cleared with a 1:2 solution of benzyl alcohol: benzyl benzoate (BABB; Sigma) before mounting between No. 1.5 coverglass. Sectioned and whole mount skin samples were imaged on a Zeiss LSM 880 confocal microscope with OPO using a 20x water objective. Image analysis was performed using FIJI.

## Cell culture

Cell culture was carried out as previously described (*Wilson et al., 2011*). Briefly, neurons from dorsal root ganglia (2–8 week old adults) or trigeminal ganglia (P0) were dissected and incubated for 10 min in 1.4 mg ml−1 Collagenase P (Roche) in Hanks calcium-free balanced salt solution, followed by incubation in 0.25% standard trypsin (vol/vol) STV versene-EDTA solution for 2 min with gentle agitation. Cells were then triturated, plated onto Poly D-Lysine coated glass coverslips and used within 20 hr. Media: MEM Eagle's with Earle's BSS medium, supplemented with 10% horse serum (vol/vol), MEM vitamins, penicillin/streptomycin and L-glutamine.

## Calcium imaging

$Ca^{2+}$ imaging experiments were carried out as previously described (*Wilson et al., 2011*). Cells were loaded for 60 min at room temperature with 10 µM Fura-2AM supplemented with 0.01% Pluronic F-127 (wt/vol, Life Technologies) in a physiological Ringer's solution containing (in mM) 140 NaCl, 5 KCl, 10 HEPES, 2 CaCl2, 2 MgCl2 and 10 D-(+)-glucose, pH 7.4. All chemicals were purchased from Sigma. Acquired images were displayed as the ratio of 340 nm/ 380 nm. Cells were identified as neurons by eliciting depolarization with high potassium Ringer's solution (75 mM) at the end of each experiment. Responding neurons were defined as those having a > 15% increase from baseline ratio. Image analysis and statistics were performed using automated routines in Igor Pro (WaveMetrics). Fura-2 ratios were normalized to the baseline ratio F340/F380 = (Ratio)/(Ratio t = 0).

## In vitro electrophysiology

Electrophysiological experiments were carried out as previously described (*Wilson et al., 2011*). Briefly, recordings were collected at 5 kHz and filtered at 2 kHz (Axopatch 200B, pClamp software). Electrode resistance ranged between 1.5–5 MΩ. Internal solution contained 140 mM KCl, 2 mM MgCl2, 1 mM EGTA, 5 mM HEPES, 1 mM Na2ATP, 100 µM GTP, and 100 µM cAMP (pH 7.4). Bath solution was physiological Ringer's solution. The pipette potential was canceled before seal formation. Cell capacitance was canceled before whole cell voltage-clamp recordings. For mechanonociceptors experiments, only cells which were visually identified as mCherry expressing and which had a capacitance between 40–80 pF were used. Rheobase was calculated as the smallest current step required to elicit an action potential using current steps of 50 pA. M currents were measured and analyzed using standard protocols for DRG neurons reported in the literature (*Schütze et al., 2016*; *Crozier et al., 2007*; *Zheng et al., 2013*). Experiments were carried out only on cells with a series resistance of less than 30 MΩ. Analysis of electrophysiology data was performed in pClamp and IgorPro.

## Ex vivo skin-nerve electrophysiology

Touch-evoked responses in the skin were recorded after dissecting the hind limb skin and saphenous nerve from 7 to 10 week old mice, according to published methods (*Wellnitz et al., 2010*; *Maksimovic et al., 2014*). The skin was placed epidermis-side-up in a custom chamber and perfused with carbogen-buffered synthetic interstitial fluid (SIF) kept at 32°C with a temperature controller (model TC-344B, Warner Instruments). The nerve was kept in mineral oil in a recording chamber, teased apart, and placed onto a gold recording electrode connected with a reference electrode to a differential amplifier (model 1800, A-M Systems). The extracellular signal was digitized using a PowerLab 8/35 board (AD Instruments) and recorded using LabChart software (AD Instruments).

For these studies, we focused on A-mechanonociceptors (AMs). To identify responses from these afferents in mutant and control genotypes, we used a mechanical search paradigm with a fine glass probe. Afferents were classified as AMs according to the following criteria: (1) conduction velocity (approximately, one to ($\leq$12 m/ s$^{-1}$), (2) medium-sized receptive fields, (3) sustained response to mechanical indentation (*Wellnitz et al., 2010*; *Koltzenburg, 1997*; *Zimmermann et al., 2009*).

Touch-sensitive afferents that did not meet these criteria were not analyzed further. Responses were classified as Adapting AMs if the ratio of mean firing rate in the dynamic phase of stimulation (first 0.2 s) to the static phase of stimulation (last 4.8 s) was greater than 2, and Non-Adapting AMs if the ratio was less than or equal to 2. Non-responders (*Figure 6F*) responded to suprathreshold mechanical stimulation with von Frey monofilaments (tip diameter <0.5 mm), but not to maximal controlled mechanical stimulation (256 mN, tip diameter 2 mm). All recordings and analyses were performed blind to genotype.

Mechanical responses were elicited with von Frey monofilaments and a force controlled custom-built mechanical stimulator. Mechanical thresholds were defined as the lowest von Frey monofilament to reliable elicit at least on action potential. Force controlled mechanical stimuli were delivered using a computer controlled, closed-loop, mechanical stimulator (Model 300C-I, Aurora Scientific, 2 mm tip diameter). Low-pass filtered, 5 s long, length control steps (square wave) simultaneously delivered with permissive force control steps (square wave) were generated using LabChart software (AD Instruments). An arbitrarily selected force step-and-hold protocol (8, 32, 4, 64, 128, 16, 256 mN) was delivered to all fibers. The period between successive displacements was 60 s.

Conduction velocity was measured by electrically stimulating identified receptive fields. Spike sorting by principal component analysis (PCA) and density based clustering, and data analysis was performed off-line with custom-made software in MATLAB. Statistics were performed in Prism.

## Flow cytometry of CFA-treated hind paws

CFA injections were performed as described above. Briefly, hindpaw skin and underlying fascia of treated and PBS-injected paws were removed from freshly euthanized mice. Skin was placed in RPMI media (Gibco) on ice before mincing with dissection scissors. Digestions were performed for 90 min at 37°C on a rotating platform in 1 mL RPMI supplemented with 1:1000 DNaseI enzyme (Roche) and one unit LiberaseTM (Roche). Skin was then filtered through 70 µm nylon mesh (Falcon), washed in RPMI, and resuspended in PBS for Aqua Live-Dead staining. Samples were then transferred to FACS buffer (PBS with 0.5% FBS and 2 mM EDTA), blocked, then surface stained with the following antibodies: cKit-Biotin, FceRI-PE, CD49b-PECy7, SiglecF-APC, SA-FITC, Ly6C-PerCP, CD11b-PB, Ly6G-BV785, CD45.2-AF700. Compensation tubes (single-stain) were prepared for each fluorophore using positive and negative control beads. A spleen from a wild-type animal was also prepared by crushing between frosted glass slides, straining through 70 µm nylon mesh, and lysing of erythrocytes in ACK (Gibco). A portion of spleen sample was heat-killed for 10 min at 65°C and stained with Aqua viability stain and set aside. The rest of the spleen was stained normally with the other skin samples. Samples were then run through a flow cytometer (BD Fortessa). Data were analyzed using FlowJo (Prism) and Microsoft Excel. Neutrophils were defined as live single cells with the following staining profile: $CD45.2^+/CD11b^+/Ly6G^+/Ly6C^+$. Inflammatory monocytes were defined as the following: $CD45.2^+/CD11b^+/Ly6G^-/Ly6C^{high}$. Total number of immune cells was reported, rather than percentage of total, since neither genotype differed significantly in total number of live cells or total number of $CD45.2^+$ immune cells.

## Statistical analyses

All statistical analyses, except for skin nerve data (see above), were performed using IgorPro software or Microsoft Excel. Values are reported as the mean ±SEM where multiple independent experiments are pooled and reported (for whole cell electrophysiology), and mean ±SD where one experiment was performed with multiple wells (for calcium imaging) or mice (for behavior). For comparison between two groups, Student's unpaired 2-tailed t-test was used. A paired t-test was employed only for measurements within the same biological replicate and after a given treatment. For single-point comparison between >2 groups, a one-way ANOVA followed by appropriate post hoc test was used, depending on comparison. For the time course comparison between two groups, 2-way ANOVA was used and single comparison p-values were derived using Tukey's HSD or appropriate statistical test, depending on comparison. Number of mice or samples required to attain significance was not calculated beforehand, and where multiple statistical tests were performed, a Bonferroni correction was applied. In figure legends, significance was labeled as: n.s., not significant, $p \geq 0.05$; *$p<0.05$; **$p<0.01$; ***$p<0.001$.

## Acknowledgements

We thank Z Rifi (UC Berkeley) for assistance with scoring itch behavior, R P Dalton (UC Berkeley) for assistance with confocal microscopy, and P Lishko and M Miller (UC Berkeley) for advice on lipid stability and usage. We would also like to thank R Clary (Columbia) for whole mount skin staining protocols, B Jenkins (Columbia) for advice on the immunohistochemistry experiments, and D Julius (UC San Francisco) for the gift of Hm1a spider toxin. Confocal imaging experiments were conducted at the CRL Molecular Imaging Center, supported by the Helen Wills Neuroscience Institute (UC Berkeley). We would like to thank H Aaron and J Lee for their microscopy training and assistance. We are grateful to all members of the DMB Laboratory (UC Berkeley) for constructive feedback and criticism. The National Institutes of Health grants NS077224 and NS098097 (to DMB and RB), AR059385 (to DMB), AR051219 (to EAL), NS105449 and GM007367 (to BUH), and NS063307 (to the Neurobiology Course at the Marine Biological Laboratory); and a Howard Hughes Medical Institute Faculty Scholars grant (to DMB) supported this work.

## Additional information

### Funding

| Funder | Grant reference number | Author |
|---|---|---|
| National Institute of Neurological Disorders and Stroke | NS077224 | Rachel B Brem<br>Diana M Bautista |
| National Institute of Arthritis and Musculoskeletal and Skin Diseases | AR059385 | Diana M Bautista |
| National Institute of Arthritis and Musculoskeletal and Skin Diseases | AR051219 | Ellen A Lumpkin |
| National Institute of Neurological Disorders and Stroke | NS105449 | Benjamin U Hoffman |
| National Institute of General Medical Sciences | GM007367 | Benjamin U Hoffman |
| Howard Hughes Medical Institute | Faculty Scholar Award | Diana M Bautista |
| National Institute of Neurological Disorders and Stroke | NS098097 | Rachel B Brem<br>Diana M Bautista |

The funders had no role in study design, data collection and interpretation, or the decision to submit the work for publication.

### Author contributions

Rose Z Hill, Conceptualization, Data curation, Formal analysis, Investigation, Visualization, Methodology, Writing—original draft, Project administration, Writing—review and editing, Performed behavior, immunostaining, whole cell electrophysiology, calcium imaging, and ISH; Benjamin U Hoffman, Data curation, Formal analysis, Investigation, Visualization, Methodology, Writing—review and editing, Performed ex vivo recordings; Takeshi Morita, Data curation, Investigation, Writing—review and editing, Performed ISH experiments; Stephanie M Campos, Investigation, Writing—review and editing, Performed ex vivo recordings; Ellen A Lumpkin, Data curation, Formal analysis, Supervision, Funding acquisition, Investigation, Project administration, Writing—review and editing; Rachel B Brem, Conceptualization, Funding acquisition, Writing—original draft, Project administration, Writing—review and editing; Diana M Bautista, Conceptualization, Supervision, Funding acquisition, Investigation, Writing—original draft, Project administration, Writing—review and editing

### Author ORCIDs

Rose Z Hill ⓘD https://orcid.org/0000-0001-9558-6400
Benjamin U Hoffman ⓘD https://orcid.org/0000-0002-9831-4061

Takeshi Morita (iD) https://orcid.org/0000-0002-8570-6744
Diana M Bautista (iD) http://orcid.org/0000-0002-6809-8951

## Ethics

Animal experimentation: All experiments were performed under the policies and recommendations of the International Association for the Study of Pain and approved by the University of California, Berkeley Animal Care and Use Committee (Protocol Number: AUP-2017-02-9550).

## Decision letter and Author response

Decision letter https://doi.org/10.7554/eLife.33285.025
Author response https://doi.org/10.7554/eLife.33285.026

## Additional files

### Supplementary files

• Supplementary file 1. Co-ISH quantification for sectioned DRG from adult wild-type mice. Related to *Figure 3*.
DOI: https://doi.org/10.7554/eLife.33285.020

• Transparent reporting form
DOI: https://doi.org/10.7554/eLife.33285.021

### Major datasets

The following previously published datasets were used:

| Author(s) | Year | Dataset title | Dataset URL | Database, license, and accessibility information |
|---|---|---|---|---|
| Usoskin D, Furlan A, Islam S, Abdo H, Lonnerberg P, Lou D, Hjerling-Leffler J, Haeggstrom J, Kharchenko O, Kharchenko PV, Linnarsson S, Ernfors P | 2015 | Unbiased classification of sensory neuron types by large-scale single-cell RNA sequencing | https://www.ncbi.nlm.nih.gov/geo/query/acc.cgi?acc=GSE59739 | Publicly available at the NCBI Gene Expression Omnibus (accession no: GSE59739) |

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
