## [Decision Letter]

[Editors’ note: this article was originally rejected after discussions between the reviewers, but the authors were invited to resubmit after an appeal against the decision.]

Thank you for submitting your work entitled "The signaling lipid sphingosine 1-phosphate regulates mechanical pain" for consideration by *eLife*. Your article has been reviewed by three peer reviewers, and the evaluation has been overseen by a Reviewing Editor and Gary Westbrook as the Senior Editor. The following individuals involved in review of your submission have agreed to reveal their identity: Bruce P Bean (Reviewer #3).

Our decision has been reached after consultation between the reviewers. Based on these discussions and the individual reviews below, we regret to inform you that your work will not be considered further for publication in *eLife*. Although the reviewers found merit in the study, a number of serious concerns were raised, and it is our editorial opinion that these concerns cannot be adequately addressed in the time frame set by *eLife* policy.

Reviewer #1:

In this manuscript, the authors characterized the role of sphingosine 1-phoshate receptor 3 (S1PR3), a candidate gene from an earlier study to screen for genes enriched in mechanosensory neurons, specifically in mechanical pain sensation. They show that (1) S1PR3 knockout mice show decreased sensitivity for noxious but not innocuous mechanical pain stimuli, (2) injection of either an S1PR3 antagonist or a kinase inhibitor that blocks S1P production also increases withdrawal thresholds, (3) S1PR3 expression in DRG partially overlaps with Npy2r, Piezo2, and Nav1.1, which marks mechosensory DRG neurons, (4) Adelta mechnosensitive fibers show decreased firing rates upon mechanical stimulation in ex vivo skin nerve preparations, and (5) S1P application blocks KCNQ2/3-mediated tail currents in dissocated DRG neuron recordings. While these studies identify an interesting new pathway that sets mechanical pain thresholds, and the authors performed a number of convincing experiments to suggest a role for S1P signaling in Adela fiber physiology, there are a few unresolved questions about the data presented.

1) Mechanical sensation usually occurs very quickly, while GPCR signaling is usually much slower. For behavior assays, the authors injected pharmacological reagents at least 30 minutes ahead of time. Thus, it doesn't seem to be very possible that S1P/S1PR3 signaling occurs when VFH was applied to the skin and modulate the mechanical sensitivity in ms. It is more likely that there is a constant baseline ongoing S1P/S1PR3 signaling that helps to set the mechanical pain threshold. How does it exactly happen? Where is the resource of this constant ligand? The authors will need to clarify their ideas/model more in the text.

2) Much of the evidence for the author's model relies on a global knockout mouse. As discussed above, if S1P/S1PR3 signaling is ongoing, would a global null mouse cause developmental deficit or other nonspecific effects that may explain the phenotype? It is also unclear the cell types/location responsible for the phenotype. While the authors do show an acute behavioral effect of a local injection of a S1PR3 antagonist and a blocker of S1P production, it does not exclude that some phenotypes they observed come from developmental deficits. Their overall model would be strengthened by a more careful characterization of the mutant mice. Use of a conditional knockout could address these concerns, either by an inducible Cre to delete S1PR3 in adulthood or restricting deletion to AM fibers. If this is not feasible, the authors could better characterize the knockout mouse, such as expression of known pain genes in DRG neurons or AM fibers, peripheral and central innervations, etc., and the developmental expression of S1PR3 in DRG neurons. For example, if S1PR3 is only expressed in adult DRG neurons, this developmental concern will be eased.

3) The characterization of S1PR3 expressing cells was only conducted to show its expression in AM neurons. It is fairly clear that this gene is expressed in other populations of DRG neurons. Based on the characterization provided, less than half of S1pr3+ neurons are AM fibers, while ~25% are Trpv1+ peptidergic neurons (although this data is not shown in the figure). Do LTMR types express S1pr3? A more thorough characterization of S1pr3 expression (NFH, peripherin, IB4, and other markers) and quantification would be helpful. Why don't the KO mice display deficits in other populations of expressing neurons? Some discussion/thoughts in this direction would be nice.

Reviewer #2:

The current study has demonstrated that the bioactive lipid sphingosine 1- phosphate (S1P) and S1P Receptor 3 (S1PR3) are critical regulators of acute mechanical pain. Although the finding is largely consistent with a previous study (Camprubi-Robles et al., 2013), the authors have provided more detailed mechanisms underlying S1P on DRG neurons. They found that S1Paffected A-delta mechanonociceptor excitability by modulating of KCNQ2/3 channels.

Although the study is potentially interesting, there are some concerns need to be addressed.

The result of S1P injection not affecting mechanosensitivity (Figure 2A) is very surprising. The authors explained that the endogenous S1P is sufficient to maximally exert its effect on S1PR3-dependent mechanical pain. However, a previous study has shown that S1P injection induced a dose dependent nociceptive response (Camprubi-Robles et al., 2013). The author should do a more careful dose response study. Another experiment can be done is that co-injection of SKI II (sphingosine kinase inhibitor) with S1P to determine whether S1P increase the attenuated mechanosensitivity by SKI II. If it is true, it suggests that S1P constitutively activates S1PR3.

The same previous study (Camprubi-Robles et al., 2013) has shown that S1PR3 in situ is expressed in nearly all DRG neurons and the staining is completely gone in S1PR3 knockout DRG (Figure 6A in that study). The authors should employ a different approach to reconcile the discrepancy such as using S1PR3mcherry/+ mice.

Why did they use P0 TG of S1PR3mcherry/+ mice to do Ca^2+^ imaging in response to Hm1a (Figure 3F)? Why not use adult DRG neurons? mCherry imaging in Figure 3F is not convincing. A better image from DRG should be provided.

Only S1PR3 heterozygous mice were used in AM recording experiments. Wild type mice should be used to see the full extent of S1PR function (Figure 4).

Reviewer #3:

This manuscript reports on the involvement of the bioactive lipid sphingosine-1-phosphate (S1P) in regulating DRG neuron excitability and pain behavior acting via the S1P receptor 3 (S1PR3). The key findings are that S1PR3-/- mice have diminished sensitivity to mechanical pain (von Frey hairs), that SP1R3 mRNA is expressed in ~40-50% of all DRG neurons, and that in SP1R3-/- mice, the response to von Frey hairs is diminished in single unit recordings from skin-nerve preparations. In addition, the authors find that S1P applied to cultured DRG neurons enhances excitability, which they propose results from closure of Kv7 channels. They propose that in vivo, endogenous S1P is high enough of produce basal closure of Kv7 channels.

The manuscript has interesting results. However, it has two serious deficiencies. One is its failure to acknowledge how much the findings overlap with previously published work, most notably with Camprubi-Robles et al., 2013. This paper is cited in the manuscript but only in a negative manner, saying that the antibodies used in that paper were found in the present work to be non-specific. There is failure to acknowledge the many results in the Camprubi-Robles paper that overlap with the results in this manuscript, including the reduced sensitivity to mechanical pain in SP1R3-/- mice, and demonstration using ISH of mRNA in many DRGS neurons (which included a nice use of SP1R3-/- mice as a control, just as in the current manuscript). Also, the authors fail to mention a series of papers from the Nicol lab showing involvement of S1P and S1PRs in pain, including showing that S1P enhances excitability of DRG neurons by reducing a potassium current and also enhancing TTX-resistant sodium current (Zhang et al.,. 2006) and that this effects is diminished using siRNA targeted to S1PR1 and SIPR2 receptors (Li et al., 2015).

A second major deficiency is in the analysis of the ionic mechanism underlying the increase in excitability reported here (and in the previous papers) by S1P applied to DRG neurons. The authors attribute this to closure of Kv7 channels (M-current). The evidence for this particular mechanism is very weak. The standard protocol for M-current is recording the slowly-deactivating outward tail currents when stepping from a steady holding potential of around -40 mV or -30 mv to around -60 mV. Here, the authors measure tail currents at -80 mV following steps to +100 mV. It is very hard to interpret these currents. It does not make sense to try to measure KV7 currents repolarizing to -80 mV, because the driving force on K is so small. In any case, I could not understand in Figure 5D how the authors were making the measurements or what the currents are. There is a sequence of fast-decaying current followed by a slowly-increasing outward current. This looks nothing like a recording of M-current (which would be a monotonically slowly-decaying outward current). The only evidence for it being mediated by Kv7 channels is occlusion by 100 μm linopirdine. This is of doubtful selectivity. The standard inhibitor of M-current (Kv2/Kv3) is XE-991, which acts with an IC50 of less than 1 uM.

This attribution of the increase in excitability by S1P to Kv7 inhibition is all the more unconvincing in light of the previous work done, which is not mentioned. Zhang et al. (2006, above) found that S1P both inhibited a high-threshold voltage-activated K current (clearly not Kv7 from its voltage dependence) and also enhanced TTX-resistant sodium channels, and Camprubi-Robles found that S1P induces a large inward current at holding potential of -80 mV (clearly not closing Kv7 channels, which would already be closed at this voltage), which they attributed to activation of chloride channels. The overall impression from all the work is that multiple conductances are affected. The evidence for closure of Kv7 channels being important is probably the least strong of any of the conductances implicated.

In my opinion, the manuscript cannot be published in its present form, based on the lack of discussion (and often, even acknowledgement) of closely related previous work and the weakness of the attribution of the S1P effects exclusively to inhibition of Kv7 channels.

[Editors’ note: what now follows is the decision letter after the authors submitted for further consideration.]

Thank you for resubmitting your work entitled "The signaling lipid sphingosine 1-phosphate regulates mechanical pain" for further consideration at *eLife*. Your revised article has been favorably evaluated by Gary Westbrook (Senior editor), a Reviewing editor, and three reviewers.

The manuscript has been improved but there are some remaining issues that need to be addressed before acceptance, as outlined below. These are relatively minor and upon resubmission the changes to the manuscript will be evaluated by the Reviewing Editor.

1) The new experiments with a conventional voltage protocol for M-current are convincing, and the new panels in Figure 5 are very nice. However, now it is hard for the reader to understand Figure 5C and 5D carried over from the previous version, or what the difference in protocol was for the "tail currents" plotted in Figure 5C and 5D versus those in Figure 5E. The voltage protocols for Figure 5C and Figure 5D are not shown, and it is not explained where tail current was measured. This should be clarified in the Figure legend (based on the previous version, presumably currents were activated by steps to +80 or +100, but I was unclear then and am still unclear at what time after the repolarization to -80 mV the tail currents were measured.). The figure might now work better leaving out Figure 5C and 5D (or moving them to the Supplementary figure) but at the least the protocols and measurements need to be explained clearly.

2) It was hard to understand Figure 6E until going deep into Materials and methods section. It is not at all obvious from the traces in the hets why the top is classified as "non-adapting" and the bottom as "adapting". Both keep firing throughout the stimulus and both show much slower firing in the second half of the stimulus. In fact, the firing during the later stimulus is faster in the "non-adapting" fiber than in the "adapting" fiber, which seems confusing. One has to go to the Materials and methods section to realize that the distinction is when frequency falls by 2-fold from the initial frequency to the end, so the initial frequency must be much faster in the bottom trace than the top. It would be much clearer if the plots included a trace of frequency averaged over 200 ms intervals or something similar. If that doesn't fit well into the figure, at least the main text should explain the criterion that is being used, and maybe say something like "the top trace shows an example of a non-adapting fiber, in which the initial frequency of xx Hz fell less than 2-fold over the 5-sec stimulus, to xx Hz. The bottom trace shows an example of a non-adapting fiber, in which the initial frequency of xx Hz fell to xx HZ by the end of the stimulus."

3) Figure 5—figure supplement 1E. "(Left)% ∆ in peak instantaneous sodium current afterS1P" – one assumes that it isn't "instantaneous" current but just "peak sodium current" during the depolarization that is plotted.

4) Subsection “Endogenous S1P mediates acute mechanical pain” "that increased of S1P does not evoke…" – delete "of".

---

## [Author Response]

[Editors’ note: the author responses to the first round of peer review follow.]

Reviewer #1:In this manuscript, the authors characterized the role of sphingosine 1-phoshate receptor 3 (S1PR3), a candidate gene from an earlier study to screen for genes enriched in mechanosensory neurons, specifically in mechanical pain sensation. They show that (1) S1PR3 knockout mice show decreased sensitivity for noxious but not innocuous mechanical pain stimuli, (2) injection of either an S1PR3 antagonist or a kinase inhibitor that blocks S1P production also increases withdrawal thresholds, (3) S1PR3 expression in DRG partially overlaps with Npy2r, Piezo2, and Nav1.1, which marks mechosensory DRG neurons, (4) Adelta mechnosensitive fibers show decreased firing rates upon mechanical stimulation in ex vivo skin nerve preparations, and (5) S1P application blocks KCNQ2/3-mediated tail currents in dissocated DRG neuron recordings. While these studies identify an interesting new pathway that sets mechanical pain thresholds, and the authors performed a number of convincing experiments to suggest a role for S1P signaling in Adela fiber physiology, there are a few unresolved questions about the data presented.1) Mechanical sensation usually occurs very quickly, while GPCR signaling is usually much slower. For behavior assays, the authors injected pharmacological reagents at least 30 minutes ahead of time. Thus, it doesn't seem to be very possible that S1P/S1PR3 signaling occurs when VFH was applied to the skin and modulate the mechanical sensitivity in ms. It is more likely that there is a constant baseline ongoing S1P/S1PR3 signaling that helps to set the mechanical pain threshold.

We agree with reviewer 1 that it is unlikely that mechanical force triggers rapid S1P release, and instead, agree that our data support a model wherein S1P/S1PR3 signaling constitutively sets the mechanical threshold. Consistent with this model, the dose-dependent effects of S1P that we measured on mechanonociceptors in vitro (Figure 5—figure supplement 1F) are saturated within the reported baseline levels of S1P in tissues in vivo (~100 nM; Ramos-Perez et al. 2015; Pappu et al. 2007). In addition, we provide new data in support of this model showing that 75-200 nM S1P is sufficient to reverse the mechanical hyposensitivity triggered by endogenous S1P depletion in vivo (Figure 2E; see response to Reviewer 2, comment 2). We have amended the Results and Discussion to highlight the data in support of a constitutive model.

How does it exactly happen? Where is the resource of this constant ligand? The authors will need to clarify their ideas/model more in the text.

We agree this is an interesting question. Indeed, this is a major research topic in immunology, where the role of S1P is well established. While we can’t answer this question, we have added the following discussion regarding the cellular source of S1P:

“What is the local source of S1P in the skin that constitutively modulates mechanical pain? Even in the mature field of S1P signaling in the vascular and immune systems, the cellular source of S1P, while an intriguing question, remains unclear. All cells in the body, including somatosensory neurons, immune cells, and skin cells, express sphingosine kinases 1 and 2, which are essential for S1P production (42). Deletion of both kinases is lethal and attempts to conditionally knockout these kinases fail to completely eliminate S1P in tissues (43). While RNA seq data suggests that somatosensory neurons contain all of the enzymatic machinery required to produce and export local S1P (19, 44) future work will be needed to identify the key cell types that are important for maintaining baseline S1P levels in the skin and regulating mechanical sensitivity.”

2) Much of the evidence for the author's model relies on a global knockout mouse. As discussed above, if S1P/S1PR3 signaling is ongoing, would a global null mouse cause developmental deficit or other nonspecific effects that may explain the phenotype? It is also unclear the cell types/location responsible for the phenotype. While the authors do show an acute behavioral effect of a local injection of a S1PR3 antagonist and a blocker of S1P production, it does not exclude that some phenotypes they observed come from developmental deficits. Their overall model would be strengthened by a more careful characterization of the mutant mice. Use of a conditional knockout could address these concerns, either by an inducible Cre to delete S1PR3 in adulthood or restricting deletion to AM fibers.

We agree that conditional knockouts are best, but have made several attempts to generate these animals using a commercially-available S1PR3 “knockout-first” strain that have thus far failed. While we will pursue the creation of a new floxed line, we believe that our discovery of S1PR3 as a key mediator of mechanical pain in the global knockout is of immediate significance, given that we know little about the molecular mechanisms underlying mechanical pain, but have provided additional, new data on the global knockout to support our claims (see below).

If this is not feasible, the authors could better characterize the knockout mouse, such as expression of known pain genes in DRG neurons or AM fibers, peripheral and central innervations, etc., and the developmental expression of S1PR3 in DRG neurons. For example, if S1PR3 is only expressed in adult DRG neurons, this developmental concern will be eased.

We agree that a more extensive characterization of the global KO would better support our findings. Indeed, we have seven new experiments, which suggest normal somatosensory neuron development in knockout animals: 1) the average diameters of S1PR3 KO and wild-type DRG cells are not statistically different (Figure 3—figure supplement 1D), and neither are the diameters of *Trpv1^+^*C nociceptors (Figure 3—figure supplement 1D) or NF200^+^ myelinated neurons (Figure 3—figure supplement 1G). 2) S1PR3 KO mice are not significantly different from WT animals with respect to IB4, NF200, or Peripherin staining of sectioned DRG (Figure 3—figure supplement 1G). 3) Expression of *Trpv1* and *Trpa1* transcripts is similar in S1PR3 KO and wild-type DRG sections (Figure 3—figure supplement 1B). 4) Similar proportions of wild type and S1PR3 KO sensory neurons respond to capsaicin in ratiometric calcium imaging (Figure 3-figure supplement 1H). 5) PGP9.5, B-tubulin III, and NefH staining in S1PR3 KO skin displays expected patterns of innervation (see Results for statistical comparison). 6) S1PR3 KO and wild-type animals display similar proportions of A-beta, A-delta (including AM nociceptors), and C fibers as measured by conduction velocity in ex vivo skin-nerve recordings (Figure 6-figure supplement 1B). Thus while we see a reduction in mechanically sensitive AM fibers by controlled force stimulation (Figure 6E-F), we observe them in normal proportions based on conduction velocity and suprathreshold von Frey stimulation. 7) We show that the mCherry reporter animals display no S1PR3 expression in epidermal and dermal cells (Figure 3D-F), and single-cell RNA seq of a diverse array of mouse skin cells corroborates this lack of expression (Joost et al. 2016). In addition, we also performed a battery of behavioral tests and found no baseline defects in sensations mediated by proprioceptors, low-threshold mechanoreceptors, thermal nociceptors, or pruriceptors in the KO animals (Figure 1).

These findings in the global knockout animal suggest that baseline S1P/S1PR3 signaling is selectively required for mechanical pain and does not affect other modalities. Overall, these experiments suggest that the phenotypes we observe in S1PR3 KO are not due to generalized defects in somatosensory development or function. Beyond the somatosensory system, S1PR3 KO animals display normal immune cell distribution and recruitment in a model of inflammatory pain (Figure 7—figure supplement 1A-B).

3) The characterization of S1PR3 expressing cells was only conducted to show its expression in AM neurons. It is fairly clear that this gene is expressed in other populations of DRG neurons. Based on the characterization provided, less than half of S1pr3+ neurons are AM fibers, while ~25% are Trpv1+ peptidergic neurons (although this data is not shown in the figure). Do LTMR types express S1pr3? A more thorough characterization of S1pr3 expression (NFH, peripherin, IB4, and other markers) and quantification would be helpful.

We have now performed a thorough ISH characterization to better characterize *S1pr3*-expressing cell types, including the use of new markers. These data show there are two main populations of *S1pr3^+^* neurons: *Scn1a+* mechanonociceptors (39.9% of *S1pr3^+^),* and *Trpv1^+^* and/or *Trpa1^+^* (67.1%) thermal nociceptors (Figure 3A-B). Staining of sectioned DRG, which is now included, shows that S1PR3 overlaps with Peripherin-expressing small-diameter neurons and IB4^+^ nociceptors as well as NF200^+^ medium-large myelinated neurons (Figure 3C,F).

Our whole mount and sectioned skin staining suggests that S1PR3 is not expressed in Merkel afferents or NefH^+^ hair follicle nerve endings (Figure 3D-E), which comprise subsets of LTMRs, but is rather expressed in NefH^+^ and NefH^-^free nerve endings, which comprise AMs and C nociceptors, respectively.

Why don't the KO mice display deficits in other populations of expressing neurons? Some discussion/thoughts in this direction would be nice.

Our original manuscript dealt only with AMs and mechanical pain; however, we have also carried out an extensive set of experiments investigating mechanisms of S1P signaling in thermal pain. Consistent with previous studies (Camprubi-Robles et al. 2013) and our data showing expression of S1PR3 in thermal nociceptors, we see that elevated S1P triggers spontaneous pain that is absent in S1PR3 KO animals (Figure 2C-D), and S1P triggers excitation of capsaicin-sensitive nociceptors (Figure 4B-E). Importantly, we also go beyond these previous studies to show that elevated S1P and S1PR3 activity promote heat hypersensitivity in the CFA model (Figure 6A, C).

Reviewer #2:The current study has demonstrated that the bioactive lipid sphingosine 1- phosphate (S1P) and S1P Receptor 3 (S1PR3) are critical regulators of acute mechanical pain. Although the finding is largely consistent with a previous study (Camprubi-Robles et al., 2013), the authors have provided more detailed mechanisms underlying S1P on DRG neurons. They found that S1Paffected A-delta mechanonociceptor excitability by modulating of KCNQ2/3 channels.Although the study is potentially interesting, there are some concerns need to be addressed.The result of S1P injection not affecting mechanosensitivity (Figure 2A) is very surprising. The authors explained that the endogenous S1P is sufficient to maximally exert its effect on S1PR3-dependent mechanical pain. However, a previous study has shown that S1P injection induced a dose dependent nociceptive response (Camprubi-Robles et al., 2013). The author should do a more careful dose response study.

Our previous manuscript focused solely on the effects of S1P signaling on mechanical pain, which is saturated by baseline levels of S1P (~200 nM). However, consistent with Camprubi-Robles et al., that reported spontaneous pain to 500 µM S1P, our dose response study shows that elevated S1P (2 µM or higher) triggers spontaneous pain in the cheek model (Figure 2C). While this other study did not measure mechanical or thermal hypersensitivity after S1P injection, our new data now show that the spontaneous pain behaviors induced by elevated S1P are due to heat and not mechanical hypersensitivity and that hypersensitivity is dependent on S1PR3 (Figure 2A-B, D). Overall, our data reveal two key, novel findings. First, S1P/S1PR3 sets baseline mechanical pain thresholds. Second, elevated S1P levels, such as those produced during inflammation or disease, promote heat hypersensitivity, but not mechanical hypersensitivity.

Another experiment can be done is that co-injection of SKI II (sphingosine kinase inhibitor) with S1P to determine whether S1P increase the attenuated mechanosensitivity by SKI II. If it is true, it suggests that S1P constitutively activates S1PR3.

We thank the reviewer for this suggestion. We have performed this experimentin vivo and found that injection of S1P does indeed reverse SKI II-induced mechanical hyposensitivity and observe a maximal effect with 200 nM S1P. This finding that exogenous S1P is sufficient to reverse the mechanical hyposensitivity triggered by the depletion of endogenous S1P supports a model whereby S1P constitutively regulates mechanical pain.

The same previous study (Camprubi-Robles et al., 2013) has shown that S1PR3 in situ is expressed in nearly all DRG neurons and the staining is completely gone in S1PR3 knockout DRG (Figure 6A in that study). The authors should employ a different approach to reconcile the discrepancy such as using S1PR3mcherry/+ mice.

We have now performed a thorough ISH characterization to better characterize *S1pr3*-expressing cell types, including the use of new markers. These data show there are two main populations of *S1pr3^+^* neurons: *Scn1a^+^* mechanonociceptors (39.9% of *S1pr3^+^),* and *Trpv1^+^* and/or *Trpa1^+^* (67.1%) thermal nociceptors (Figure 3A-B). Staining of sectioned DRG, which is now included, shows that S1PR3 overlaps with Peripherin-expressing small-diameter neurons and IB4+ nociceptors as well as NF200+ medium-large myelinated neurons (Figure 3C,F). Our staining data complement findings from a recent single-cell RNA-seq study showing expression of S1pr3 in two populations comprising a subset of DRG neurons: the NF200+ myelinated mechanoreceptors and peptidergic nociceptors (Usoskin et al. 2015).

Unfortunately, we cannot make direct comparisons of our data to theirs, as they did not examine co-localization with other markers. And although they conclude that *S1pr3* is expressed by “virtually all” somatosensory neurons, they did not quantify *S1pr3* expression in wild type ganglia. Regardless, we now provide extensive new data, which, combined with our previous data, supports selective expression of S1PR3 primarily in two neuronal subsets: mechanonociceptors and thermal nociceptors.

Why did they use P0 TG of S1PR3mcherry/+ mice to do Ca^2+^ imaging in response to Hm1a (Figure 3F)? Why not use adult DRG neurons? mCherry imaging in Figure 3F is not convincing. A better image from DRG should be provided.

Our use of P0 TG was based on the original characterization of Hm1a by Osteen and colleagues (Osteen et al. 2016) who reported that PGE2 sensitization is required to see robust and sustained Hm1a-evoked calcium influx in adult DRG neurons, but that Hm1a robustly activates naive P0 trigeminal ganglion neurons (due to faster inactivation kinetics in the adult). Thus, after consultation with Dr. Julius, we used P0 neurons to avoid the added complication of pre-sensitizing neurons that could change excitability. We have added the following passages to the results (p.8) and figure caption (Figure 4), and also provide a clearer, more representative image to replace the previous version (Figure 4F):

“To this end, we asked whether the spider toxin Hm1a, a selective activator of AM nociceptors (28), triggers calcium influx in S1PR3-expressing trigeminal neurons. Indeed, we found that 44.2 ± 15.1% of Hm1aresponsive neurons expressed mCherry (Figure 4F), consistent with our staining showing expression of *S1pr3* in AM nociceptors and the role of Hm1a-responsive neurons in mediating mechanical pain in vivo (28).”

“Figure 4F. (Left) Fura-2 AM calcium imaging after addition of 500 nM Hm1a in *S1pr3^mCherry/+^* P0 TG neurons. TG neurons were used in calcium imaging instead of adult DRG neurons because they maximally respond to Hm1a without prior PGE2 sensitization, due to slower sodium channel inactivation kinetics. Right-hand image indicates mCherry fluorescence. (Right)% of Hm1a-responsive P0 TG neurons that are mCherry+ (N = 1 animal, 1230 total neurons).”

Only S1PR3 heterozygous mice were used in AM recording experiments. Wild type mice should be used to see the full extent of S1PR function (Figure 4).

We now include recordings from S1PR3 WT AM fibers, which do not significantly differ from S1PR3 HET fibers in mechanically-evoked responses to controlled force stimulation (Figure 6-figure supplement 1A), and cite a number of other studies on AM fibers that report similar force-response curves for their wild-type recordings (Osteen et al. 2016, Garrison et al. 2015, Smith et al. 2013, Hillery et al. 2011, Kwan et al. 2009, McIlwrath et al. 2007). This now allows for the direct comparison of the effects of S1PR3 signaling across genotypes.

Reviewer #3:This manuscript reports on the involvement of the bioactive lipid sphingosine-1-phosphate (S1P) in regulating DRG neuron excitability and pain behavior acting via the S1P receptor 3 (S1PR3). The key findings are that S1PR3-/- mice have diminished sensitivity to mechanical pain (von Frey hairs), that SP1R3 mRNA is expressed in ~40-50% of all DRG neurons, and that in SP1R3-/- mice, the response to von Frey hairs is diminished in single unit recordings from skin-nerve preparations. In addition, the authors find that S1P applied to cultured DRG neurons enhances excitability, which they propose results from closure of Kv7 channels. They propose that in vivo, endogenous S1P is high enough of produce basal closure of Kv7 channels.1) The manuscript has interesting results. However, it has two serious deficiencies. One is its failure to acknowledge how much the findings overlap with previously published work, most notably with Camprubi-Robles et al., 2013. This paper is cited in the manuscript but only in a negative manner, saying that the antibodies used in that paper were found in the present work to be non-specific. There is failure to acknowledge the many results in the Camprubi-Robles paper that overlap with the results in this manuscript, including the reduced sensitivity to mechanical pain in SP1R3-/- mice…

We agree that it is essential to put our data in the context of this study, especially since the scope of our studies is quite different: a distinction that was not highlighted adequately in our original manuscript that only discussed AMs and mechanical pain. Our previous manuscript focused solely on the effects of S1P signaling on mechanical pain, which is saturated by baseline levels of S1P (~200 nM). However, consistent with Camprubi-Robles et al., that reported spontaneous pain to 500 µM S1P, we also observe that elevated S1P (2 µM or higher) triggers spontaneous pain in the cheek model (Figure 2C). While this other study did not measure mechanical or heat hypersensitivity after S1P injection, our new data now show that the spontaneous pain behaviors induced by elevated S1P are due to heat and not mechanical hypersensitivity and that hypersensitivity is dependent on S1PR3 (Figure 2A-B, D). Overall, our data reveal two key, novel findings. First, S1P/S1PR3 sets baseline mechanical pain thresholds. Second, elevated S1P levels, such as those produced during inflammation or disease, promote heat hypersensitivity, but not mechanical hypersensitivity.

In addition to clarifying our results in the context of these previous findings throughout the revised manuscript, we have also added a significant section to the discussion that puts our results in context with past studies:

“Previous studies of S1P signaling in DRG neurons focused on S1P-evoked excitation of small diameter and/or capsaicin-sensitive neurons, and pain behaviors triggered by elevated S1P. […] Our observation that S1PR3 KO animals display normal immune cell infiltration and develop mechanical hypersensitivity after CFA is consistent with previous studies showing distinct mechanisms of inflammatory thermal and mechanical hypersensitivity.”

…[D]emonstration using ISH of mRNA in many DRGS neurons (which included a nice use of SP1R3-/- mice as a control, just as in the current manuscript).

Unfortunately, we cannot make direct comparisons of our data to theirs, as they did not examine co-localization with other markers. And although they conclude that *S1pr3* is expressed by “virtually all” somatosensory neurons, they did not quantify *S1pr3* expression in wild type ganglia. Regardless, we now provide extensive new data, which, combined with our previous data, supports selective expression of S1PR3 in two neuronal subsets: mechanonociceptors and thermal nociceptors.

We have now performed a thorough ISH characterization to better characterize S1pr3-expressing cell types, including the use of new markers. These data show there are two main populations of *S1pr3^+^* neurons: *Scn1a^+^* mechanonociceptors (39.9% of *S1pr3^+^*), and *Trpv1^+^* and/or *Trpa1^+^* (67.1%) thermal nociceptors (Figure 3A-B). Staining of sectioned DRG, which is now included, shows that S1PR3 overlaps with Peripherin-expressing small-diameter neurons and IB4^+^ nociceptors as well as NF200+ medium-large myelinated neurons (Figure 3C,F).

Our whole mount and sectioned skin staining suggests that S1PR3 is not expressed in Merkel afferents or NefH^+^ hair follicle nerve endings (Figure 3D-E), which comprise subsets of LTMRs, but is rather expressed in NefH^+^ and NefH^free^ nerve endings, which comprise AMs and C nociceptors, respectively.

2) Also, the authors fail to mention a series of papers from the Nicol lab showing involvement of S1P and S1PRs in pain, including showing that S1P enhances excitability of DRG neurons by reducing a potassium current and also enhancing TTX-resistant sodium current (Zhang et al., 2006) and that this effect is diminished using siRNA targeted to S1PR1 and SIPR2 receptors (Li et al., 2015).

Our current work represents the first analysis of S1P signaling in AM nociceptors, and is distinct from work performed in these previous papers, for a number of reasons. All previous studies of S1P signaling in DRG neurons, including those cited above, focused on small diameter and/or capsaicin-sensitive neurons. While our new data affirms the effects in thermal nociceptors observed by others (Figure 4B-E), our manuscript highlights a novel effect of S1P in modulation of rheobase and M current in mechanonociceptors. Moreover, the effects of S1P on excitability of mechanonociceptors are completely lost in the S1PR3 knockout (Figure 5A). We have added the following passages to the Results and Discussion to clarify this:

“We next interrogated the molecular mechanism by which S1P signaling in AM nociceptors may regulate mechanical pain. […] Additionally, S1P significantly reduced slow, voltage-dependent tail current amplitudes (Figure 5D; Figure

5E (top)) in an S1PR3-dependent manner (Figure 5D, center).”

“Previous studies of S1P signaling in DRG neurons focused on S1P-evoked excitation of small diameter and/or capsaicin-sensitive neurons, and pain behaviors triggered by high S1P. While our new data affirms the effects S1P in thermal nociceptors observed by others, our manuscript highlights a novel effect of S1P in modulation of rheobase and KCNQ2/3 currents in mechanonociceptors.”

3) A second major deficiency is in the analysis of the ionic mechanism underlying the increase in excitability reported here (and in the previous papers) by S1P applied to DRG neurons. The authors attribute this to closure of Kv7 channels (M-current). The evidence for this particular mechanism is very weak. The standard protocol for M-current is recording the slowly-deactivating outward tail currents when stepping from a steady holding potential of around -40 mV or -30 mv to around -60 mV. Here, the authors measure tail currents at -80 mV following steps to +100 mV. It is very hard to interpret these currents. It does not make sense to try to measure KV7 currents repolarizing to -80 mV, because the driving force on K is so small. In any case, I could not understand in Figure 5D how the authors were making the measurements or what the currents are. There is a sequence of fast-decaying current followed by a slowly-increasing outward current. This looks nothing like a recording of M-current (which would be a monotonically slowly-decaying outward current). The only evidence for it being mediated by Kv7 channels is occlusion by 100 μm linopirdine. This is of doubtful selectivity. The standard inhibitor of M-current (Kv2/Kv3) is XE-991, which acts with an IC50 of less than 1 uM.This attribution of the increase in excitability by S1P to Kv7 inhibition is all the more unconvincing in light of the previous work done, which is not mentioned. Zhang et al. (2006, above) found that S1P both inhibited a high-threshold voltage-activated K current (clearly not Kv7 from its voltage dependence) and also enhanced TTX-resistant sodium channels, and Camprubi-Robles found that S1P induces a large inward current at holding potential of -80 mV (clearly not closing Kv7 channels, which would already be closed at this voltage), which they attributed to activation of chloride channels. The overall impression from all the work is that multiple conductances are affected. The evidence for closure of Kv7 channels being important is probably the least strong of any of the conductances implicated.

We have performed new experiments that address these concerns. While we originally reported results using linopirdine and +100 to -80 mV voltage steps, we also performed experiments with the suggested M current recording protocol (-40 to -60 mV steps). Using this protocol, the effects of S1P on M current were completely occluded by XE-991 (Figure 5E). These results dovetail with previous studies showing that KCNQ2/3 channels mediate M currents in cultured A-delta neurons, which include AM nociceptors and D-hairs (Schutze et al. 2016; Crozier et al. 2013), and support our model wherein S1P/S1PR3 constitutively modulate M currents in AM nociceptors.

In regard to the effects described by the aforementioned studies on thermal nociceptors (Figure 4B-E), we do not observe S1P-evoked excitation of or changes in sodium or steady-state potassium currents in mechanonociceptors (Figure 5-figure supplement 1C-E). These additional concerns are addressed above (see response to Reviewer 3 comment 1). See also our added results regarding S1P-evoked inhibition of KCNQ2/3 currents in mechanonociceptors:

“As tail currents in Aδ neurons are primarily mediated by KCNQ2/3 potassium channels (30, 31), we postulated that S1P may alter tail currents through modulation of these channels. […] In summary, our electrophysiological and behavioral observations support a model in which baseline S1P/S1PR3 signaling governs mechanical pain thresholds through modulation of KCNQ2/3 channel activity in AM neurons (Figure 8).”

[Editors’ note: the author responses to the re-review follow.]

1) The new experiments with a conventional voltage protocol for M-current are convincing, and the new panels in Figure 5 are very nice. However, now it is hard for the reader to understand Figure 5C and 5D carried over from the previous version, or what the difference in protocol was for the "tail currents" plotted in Figure 5C and D versus those in Figure 5E. The voltage protocols for Figure 5C and Figure D are not shown, and it is not explained where tail current was measured. This should be clarified in the Figure legend (based on the previous version, presumably currents were activated by steps to +80 or +100, but I was unclear then and am still unclear at what time after the repolarization to -80 mV the tail currents were measured.). The figure might now work better leaving out Figure C and D (or moving them to the Supplementary figure) but at the least the protocols and measurements need to be explained clearly.

We thank the reviewers for the suggestion to clarify Figure 5. For Figure 5C, used an IV protocol whereby we recorded the current from -100 to +80mV in incremental 20mV steps, before and after S1P application. We have now clarified this in the figure legend as follows:

“C. The S1P-sensitive current is carried by potassium. The current-voltage relationship was determined by subtraction of the post-S1P current from the pre-S1P current and reverses at -60.125 mV; N = 6 cells. Data were fitted with a Boltzmann equation. Pre- and post-S1P currents were measured at the indicated voltage (-100mV to +80mV, 20mV increments) following a +100 mV step (100 ms). Current was quantified using the peak absolute value of the slowly-deactivating current 0-10 ms after stepping to indicated voltage. Unless indicated otherwise, all error bars represent mean ± SEOM. D. (Graphic, top) Averaged current traces of a single mCherry+ neuron in whole cell voltage clamp recording comparing tail currents (∆_I tail_) pre- and post-S1P using indicated voltage step protocol. (graphic, bottom) Averaged current traces of a single mCherry+ neuron in whole cell voltage clamp recording with XE991 treatment. Holding phase (-40 mV, 150 ms) was truncated in traces. (Left graph)% ∆ in outward tail current (average +/- SD after indicated treatments (1 µM S1P, 3 µM XE 991, or both) for S1pr3^mCherry/+^ medium-diameter neurons; (p = 0.58; one-way ANOVA; n = 6, 8, 14 cells) using protocol depicted at right. (Right graph)% ∆ in inward tail current after indicated treatments (LINO = 100 µM linopirdine) for S1pr3^mCherry/+^ medium-diameter neurons; (p = 0.47; two-tailed paired t-test; N = 12 cells).”

As suggested, we have also moved Figure 5D to Figure 5—figure supplement 1F. The supplemental figure legend now reads:

“F.% ∆ in inward tail current (∆_I tail_) after S1P or 1% DMSO vehicle application for S1pr3^mCherry/+^ and KO medium-diameter neurons using a pre-pulse stimulation of +80 mV followed by a step to -80 mV, where (∆_I tail_) was calculated by subtracting the steady-state current from the absolute peak of the slowly-deactivating current at -80 mV (p = 0.014; one-way ANOVA; N = 10, 13, 10 cells). Tukey Kramer post hoc p-values indicated on graph.”

2) It was hard to understand Figure 6E until going deep into Materials and methods section. It is not at all obvious from the traces in the hets why the top is classified as "non-adapting" and the bottom as "adapting". Both keep firing throughout the stimulus and both show much slower firing in the second half of the stimulus. In fact, the firing during the later stimulus is faster in the "non-adapting" fiber than in the "adapting" fiber, which seems confusing. One has to go to the Methods to realize that the distinction is when frequency falls by 2-fold from the initial frequency to the end, so the initial frequency must be much faster in the bottom trace than the top. It would be much clearer if the plots included a trace of frequency averaged over 200 ms intervals or something similar. If that doesn't fit well into the figure, at least the main text should explain the criterion that is being used, and maybe say something like "the top trace shows an example of a non-adapting fiber, in which the initial frequency of xx Hz fell less than 2-fold over the 5-sec stimulus, to xx Hz. The bottom trace shows an example of a non-adapting fiber, in which the initial frequency of xx Hz fell to xx HZ by the end of the stimulus."

We thank the reviewers for the suggestion to clarify Figure 6E and the corresponding main text. We have updated the figure to include the binned instantaneous firing-rate frequency and the caption now reads:

“E. Representative traces and binned instantaneous firing frequencies (IFF; 200-ms bins) of Non-Adapting and Adapting AMs in response to force controlled stimulation (256 mN, top) for S1PR3 HET and KO mice; blue regions, dynamic phase of stimulation (200-ms).”

We also modified the associated main text as follows (changes underlined):

“A recent study reported that A-nociceptors are composed of two genetically distinct neuronal populations that differ in conduction velocity and in adaptation properties^5^ (“Adapting AM” versus “Non-adapting AM”). We next asked whether loss of S1PR3 signaling altered these AM subtypes. Adapting AM fibers responded more vigorously to dynamic (ramp) stimuli than static (hold) stimuli, and displayed a mean dynamic firing frequency at least twofold greater than their static firing frequency^5^ (Figure 6E, upper traces). By contrast, Non-adapting AM fibers often showed bursting during static stimulation, which resulted in similar firing rates during dynamic and static stimulation (Figure 6E, lower traces). S1PR3 KO animals displayed a significantly lower proportion of Adapting AM nociceptors compared with littermate controls (Figure 6F). Additionally, we observed an increase in S1PR3 KO AM fibers that were unresponsive to controlled force stimulation (Figure 6F). These “non-responders” only fired action potentials to high-pressure stimuli with a blunt glass probe or to suprathreshold stimulation with von Frey filaments (see Materials and methods section). The Non-Adapting AMs, and the few remaining mechanosensitive Adapting AMs in the S1PR3 KO displayed similar firing frequencies over both the dynamic and static phases of force application to control fibers (Figure 6—figure supplement 1C). This suggests that decreased mechanosensitivity of the Adapting AM population accounts for the significant reduction in force-firing relations observed at the population level in S1PR3 KO AMs (Figure 6A). We conclude that S1PR3 is an essential regulator of both mechanical threshold and sensitivity in a distinct population of AM nociceptors.”

Additionally, we have included a supplemental figure (Figure 6—figure supplement 1C) to clarify the phenotype observed in S1PR3 KO Adapting AMs that is described by the new main text. The figure caption reads:

“C. Mean firing rates during dynamic (ramp) and static (hold) stimulation for S1PR3 HET and S1PR3 KO recordings (left, Adapting AMs; right, Non-Adapting AMs; see Figure 6E-F for experimental details). No significant differences were found between genotypes (p = 0.227, 0.490 (two-way ANOVA); bars, means). As shown in Figure 6F, the proportion of Adapting AMs was significantly lower in S1PR3 KO recordings compared with littermate controls.”

3) Figure 5—figure supplement 1E. "(Left)% ∆ in peak instantaneous sodium current afterS1P" – one assumes that it isn't "instantaneous" current but just "peak sodium current" during the depolarization that is plotted.

The figure legend now reads “% ∆ in peak sodium current”, as requested.

4) Subsection “Endogenous S1P mediates acute mechanical pain” "that increased of S1P does not evoke…" – delete "of".

This portion of the main text now reads “that increased S1P does not evoke…”.